# Regulation of immune receptor kinase plasma membrane nanoscale organization by a plant peptide hormone and its receptors

**Julien Gronnier[1,2]\*[†], Christina M Franck[1], Martin Stegmann[2‡], Thomas A DeFalco[1,2], Alicia Abarca[1], Michelle von Arx[1], Kai Dünser[3], Wenwei Lin[4], Zhenbiao Yang[4], Jürgen Kleine-Vehn[3], Christoph Ringli[1], Cyril Zipfel[1,2]\***

[1]Institute of Plant and Microbial Biology and Zurich-Basel Plant Science Center, University of Zurich, Zurich, Switzerland; [2]The Sainsbury Laboratory, University of East Anglia, Norwich Research Park, Norwich, United Kingdom; [3]Department of Applied Genetics and Cell Biology, University of Natural Resources and Life Sciences Vienna, Vienna, Austria; [4]FAFU-UCR Joint Center for Horticultural Biology and Metabolomics Center, Haixia, Institute of Science and Technology, Fujian Agriculture and Forestry University, Fuzhou, China

**\*For correspondence:**
julien.gronnier@zmbp.uni-tuebingen.de (JG);
cyril.zipfel@botinst.uzh.ch (CZ)

**Present address:** [†]University of Tübingen, Center for Plant Molecular Biology (ZMBP), Tübingen, Germany; [‡]Phytopathology, School of Life Sciences Weihenstephan, Technical University of Munich, Freising, Germany

**Abstract** Spatial partitioning is a propensity of biological systems orchestrating cell activities in space and time. The dynamic regulation of plasma membrane nano-environments has recently emerged as a key fundamental aspect of plant signaling, but the molecular components governing it are still mostly unclear. The receptor kinase FERONIA (FER) controls ligand-induced complex formation of the immune receptor kinase FLAGELLIN SENSING 2 (FLS2) with its co-receptor BRASSINOSTEROID-INSENSITIVE 1-ASSOCIATED KINASE 1 (BAK1), and perception of the endogenous peptide hormone RAPID ALKALANIZATION FACTOR 23 (RALF23) by FER inhibits immunity. Here, we show that FER regulates the plasma membrane nanoscale organization of FLS2 and BAK1. Our study demonstrates that akin to FER, leucine-rich repeat (LRR) extensin proteins (LRXs) contribute to RALF23 responsiveness and regulate BAK1 nanoscale organization and immune signaling. Furthermore, RALF23 perception leads to rapid modification of FLS2 and BAK1 nanoscale organization, and its inhibitory activity on immune signaling relies on FER kinase activity. Our results suggest that perception of RALF peptides by FER and LRXs actively modulates plasma membrane nanoscale organization to regulate cell surface signaling by other ligand-binding receptor kinases.

## Editor's evaluation

In elegant quantitative live-cell imaging and biochemical experiments, the authors show how activity of the plant immune signaling complex FLS2-BAK1 is affected by nanoscale mobility behaviors mediated through peptide signaling and the receptor kinase FERONIA (FER). Additionally, they are able to define separable roles for FER domains in different biological activities. The details of this work advance our understanding of plant immunity, but also provide generalizable concepts about the roles of nanoscale organization in signaling.

## Introduction

Multicellular organisms evolved sophisticated surveillance systems to monitor changes in their environment. In plants, receptor kinases (RKs) and receptor proteins (RPs) are the main ligand-binding cell-surface receptors perceiving self, non-self, and modified-self molecules (*Hohmann et al., 2017*). For example, recognition of pathogen-associated molecular patterns (PAMPs) by pattern recognition receptors (PRRs) initiates signaling events, leading to pattern-triggered immunity (PTI) (*DeFalco and Zipfel, 2021*). The *Arabidopsis thaliana* (hereafter *Arabidopsis*) leucine-rich repeat receptor kinases (LRR-RKs) FLS2 and EFR recognize the bacterial PAMPs flagellin (or its derived epitope flg22) and elongation factor-Tu (or its derived epitope elf18), respectively (*Gómez-Gómez and Boller, 2000*; *Zipfel et al., 2006*). Both FLS2 and EFR form ligand-induced complexes with the co-receptor BAK1 (a LRR-RK also referred as SERK3) to initiate immune signaling, such as the production of apoplastic reactive oxygen species (ROS), and calcium influx (*Chinchilla et al., 2007*; *Heese et al., 2007*; *Schulze et al., 2010*; *Roux, 2011*; *Sun et al., 2013*; *Thor et al., 2020*).

We previously showed that the *Catharanthus roseus* RECEPTOR-LIKE PROTEIN KINASE 1-LIKE (CrRLK1L) FERONIA (FER) and the GPI-anchored protein LORELEI-LIKE GPI-ANCHORED PROTEIN 1 (LLG1) are required for flg22-induced FLS2-BAK1 complex formation (*Stegmann et al., 2017*; *Xiao et al., 2019*). Notably, the endogenous peptide hormone RALF23 is perceived by a LLG1-FER heterocomplex, which leads to inhibition of flg22-induced FLS2-BAK1 complex formation (*Stegmann et al., 2017*; *Xiao et al., 2019*). As such, although FER and LLG1 are positive regulator of PTI, RALF23 is a negative regulator. How these components regulate FLS2-BAK1 complex formation remains however unclear.

Several members of the CrLKL1L family are involved in RALFs perception (*Haruta et al., 2014*; *Ge et al., 2017*; *Gonneau et al., 2018*; *Liu, 2021*). Among them, FER plays a pivotal role in the perception of several *Arabidopsis* RALF peptides (*Haruta et al., 2014*; *Stegmann et al., 2017*; *Gonneau et al., 2018*; *Zhao et al., 2018*; *Abarca et al., 2021*; *Liu, 2021*). In addition, cell wall-associated LEUCINE-RICH REPEAT-EXTENSINs (LRXs) proteins are also involved in CrRLK1L-regulated pathways and were shown to bind RALFs with high affinity (*Mecchia, 2017*; *Zhao et al., 2018*; *Dünser, 2019*; *Herger et al., 2020*; *Moussu, 2020*). Structural and biochemical analyses indicate that RALF binding by CrRLK1L/LLG complexes and LRXs are mutually exclusive and mechanistically distinct from each other (*Xiao et al., 2019*; *Moussu, 2020*). While CrRLK1Ls and LRXs have emerged as important RALF-regulated signaling modules, it is still unknown whether LRXs are also involved in RALF23-mediated regulation of immune signaling.

Plasma membrane lipids and proteins dynamically organize into diverse membrane domains giving rise to fluid molecular patchworks (*Gronnier et al., 2018*; *Ballweg et al., 2020*; *Jaillais and Ott, 2020*). These domains are proposed to provide dedicated biochemical and biophysical environments to ensure acute, specific, and robust signaling events (*Gronnier et al., 2019*; *Jacobson et al., 2019*). For instance, FLS2 localizes in discrete and static structures proposed to specify immune signaling (*Bücherl et al., 2017*). The cell wall is thought to impose physical constraints on the plasma membrane, limiting the diffusion of its constituents (*Feraru, 2011*; *Martinière, 2012*). Indeed, alteration of cell wall integrity leads to aberrant protein motions at the plasma membrane (*Martinière, 2012*; *McKenna, 2019*). Notably, perturbation of the cell wall affects FLS2 nanoscale organization (*McKenna, 2019*). Despite its utmost importance, it remains largely unknown how the cell wall and its integrity modulate the organization of the plasma membrane. Interestingly, both CrRLK1Ls and LRXs are proposed cell wall integrity sensors and conserved modules regulating growth, reproduction, and immunity (*Franck et al., 2018*; *Herger et al., 2019*). However, their mode of action and potential links between cell wall integrity sensing and RALF perception are still poorly understood.

Here, we show that FER regulates the plasma membrane nanoscale organization of FLS2 and BAK1. Similarly, we show that LRXs contribute to RALF23 responsiveness and regulate BAK1 nanoscale organization and immune signaling. Importantly, our work reveals an unexpected uncoupling of FER and LRX modes of action in growth and immunity. We demonstrate that RALF23 perception leads to rapid modulation of FLS2 and BAK1 nanoscale organization and that its inhibitory activity on immune signaling requires FER kinase activity. We propose that the regulation of the plasma membrane nanoscale organization by RALF23 receptors underscores their role in the formation of protein complexes and initiation of immune signaling.

## Results and discussion

### FER regulates membrane nanoscale organization of FLS2 and BAK1

We combined variable angle total internal reflection fluorescence microscopy (VA-TIRFM) and single-particle tracking to analyze the lateral mobility of FLS2-GFP in transgenic *Arabidopsis* lines. Two lines expressing FLS2-GFP under the control of its native promoter were crossed with *FER* knock-out alleles *fer-2* and *fer-4*. In line with previous reports (*Bücherl et al., 2017*; *Tran et al., 2020*), we observed that FLS2-GFP localized to laterally stable foci in wild-type (WT) (*Figure 1—video 1*). Consistently, FLS2-GFP single-particle trajectories exhibited a confined mobility behavior (*Figure 1—figure supplement 1*, *Figure 1—video 1*). Comparative analysis of the diffusion coefficient (D), which describes the diffusion properties of detected single particles (*Kusumi et al., 1993*), showed that FLS2-GFP was more mobile in *fer* mutants than in WT (*Figure 1—figure supplement 1*, *Figure 1—figure supplement 2*, and *Figure 1—video 1*). To analyze FLS2-GFP organization, we reconstructed images using a temporal averaging of FLS2-GFP fluorescence observed across VA-TIRFM time series. Furthermore, individual image sections were subjected to kymograph analysis. Using this approach, we found that FLS2-GFP fluorescence was maintained into well-defined and static structures in WT, while it appeared more disperse and more mobile in both *fer* mutants (*Figure 1A and B*, *Figure 1—figure supplement 2*). To substantiate these observations, we used the previously established spatial clustering index (SCI), which describes protein lateral organization (*Gronnier et al., 2017*; *Tran et al., 2020*). As expected, SCI of FLS2-GFP was lower in *fer-4* than in WT (*Figure 1C*), indicating disturbance in FLS2-GFP lateral organization.

In *Medicago truncatula* and yeast, alteration of nanodomain localization has been linked to impaired protein accumulation at the plasma membrane due to increased protein endocytosis (*Grossmann et al., 2008*; *Liang et al., 2018*). To inquire for a potential defect in FLS2 plasma membrane accumulation, we observed subcellular localization of FLS2-GFP using confocal microscopy. The analysis revealed a decrease in FLS2-GFP accumulation in *fer* mutants (*Figure 1—figure supplement 3*). Whether the proposed role of FER in regulating endocytosis (*Yu et al., 2020*) accounts for this defect is unknown. Altogether, these results show that *FER* is genetically required to control FLS2-GFP nanoscale organization and accumulation at the plasma membrane.

To further characterize the impact of *FER* loss of function in RK organization, we analyzed the behavior of BAK1-mCherry at the plasma membrane. Fluorescence recovery after photobleaching experiments previously suggested that the vast majority of BAK1 molecules are mobile (*Hutten et al., 2017*). Consistent with this result, BAK1-mCherry was more mobile than FLS2-GFP in the WT (*Figure 1—video 2*). Given that BAK1 is a common co-receptor for multiple LRR-RK signaling pathways (*Hohmann et al., 2017*), we hypothesized that BAK1 might dynamically associate with various pre-formed signaling platforms, such as FLS2 nanodomains (*Figure 1*, *Bücherl et al., 2017*). Under our experimental conditions, we were not able to perform high-quality single-particle tracking analysis for BAK1-mCherry (*Figure 1—video 2*, see Materials and methods section). However, visual inspection of particles behavior suggested that BAK1-mCherry was less mobile in *fer-4* than in WT (*Figure 1—video 2*). Accordingly, reconstructed VA-TIRFM images and kymographs showed that BAK1-mCherry fluorescence was more structured and static in *fer-4* than in WT (*Figure 1F*). Furthermore, we observed an increase of BAK1-mCherry SCI in *fer-4* (*Figure 1G*). Confocal microscopy analysis did not reveal significant differences in BAK1-mCherry plasma membrane accumulation between *fer-4* and WT (*Figure 2—figure supplement 1*). Altogether, these data show that loss of *FER* perturbs FLS2 and BAK1 nanoscale organization, albeit in an opposite manner (*Figure 1D and H*). Previous reports have similarly shown that altering the composition of the cell wall can lead to opposed effects on the mobility of different proteins. For instance, inhibition of cellulose synthesis increases the mobility of HYPERSENSITIVE-INDUCED REACTION 1 (*Daněk et al., 2020*) but limits the mobility of LOW-TEMPERATURE-INDUCED PROTEIN 6B (*Martinière, 2012*; *Daněk et al., 2020*). Modification of pectin methyl esterification status of the cell wall increases the mobility of FLS2 (*McKenna, 2019*) but decreases the mobility of FLOTILIN 2 (*Daněk et al., 2020*). Collectively, these observations suggest that various membrane environments are differentially regulated by the cell wall and the proposed cell wall integrity sensor FER.

### LRX3, LRX4, and LRX5 regulate BAK1 nanoscale organization and PTI signaling

LRXs are dimeric, cell wall-localized, high-affinity RALF-binding proteins suggested to monitor cell wall integrity in growth and reproduction (*Baumberger et al., 2001*; *Mecchia, 2017*; *Dünser, 2019*;

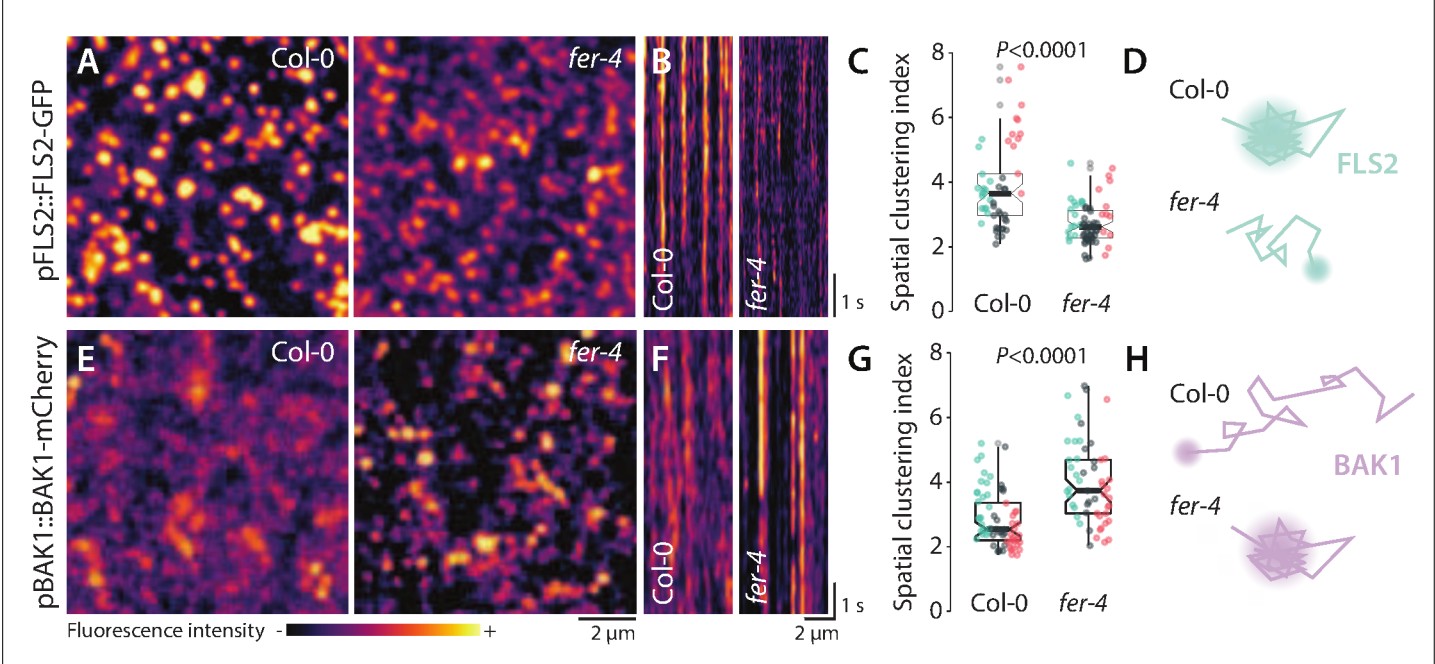

**Figure 1.** FER regulates the nanoscale organization of FLS2-GFP and BAK1-mCherry. (**A**, **E**) FLS2-GFP and BAK1-mCherry nanodomain organization. Pictures are maximum projection of 20 variable angle total internal reflection fluorescence microscopy (VA-TIRFM) images obtained at 5 frames per second for FLS2-GFP (**A**) and 10 VA-TIRFM images obtained at 2.5 frames per second for BAK1-mCherry (**E**) in Col-0 and *fer-4* cotyledon epidermal cells. (**B**, **F**) Representative kymograph showing lateral organization of FLS2-GFP (**B**) and BAK1-mCherry (**F**) overtime in Col-0 and *fer-4*. (**C**, **G**) Quantification of FLS2-GFP (**C**) and BAK1-mCherry (**G**) spatial clustering index. Graphs are notched box plots, scattered data points show measurements, colors indicate independent experiments, n = 16 cells for Col-0/pFLS2::FLS2-GFP; n = 31 cells for *fer-4*/pFLS2::FLS2-GFP, n = 23 cells for Col-0/pBAK1::BAK1-mCherry, n = 18 cells for *fer-4*/pBAK1::BAK1-mCherry. p-Values report two-tailed nonparametric Mann–Whitney test. (**D**, **H**) Graphical illustrations summarizing our observations for FLS2-GFP (**D**) and BAK1-mCherry (**H**) nanoscale dynamics.

The online version of this article includes the following video, source data, and figure supplement(s) for figure 1:

**Source data 1.** Source data points for the graphs in *Figure 1C and G*.

**Figure supplement 1.** Analysis of FLS2-GFP single-particle dynamics in *fer-4*.

**Figure supplement 1—source data 1.** Source data points for the graph in *Figure 1—figure supplement 1*.

**Figure supplement 2.** Analysis of FLS2-GFP organization and dynamics in *fer-2*.

**Figure supplement 2—source data 1.** Source data points for the graphs in *Figure 1—figure supplement 2B and D*.

**Figure supplement 3.** FLS2-GFP accumulation at the PM is altered in *fer* mutants.

**Figure supplement 3—source data 1.** Source data points for the graphs in *Figure 1—figure supplement 3B and D*.

**Figure 1—video 1.** Variable angle total internal reflection fluorescence microscopy (VA-TIRFM) imaging of FLS2-GFP in Col-0 and *fer-4*.
https://elifesciences.org/articles/74162/figures#fig1video1

**Figure 1—video 2.** Variable angle total internal reflection fluorescence microscopy (VA-TIRFM) imaging of BAK1-mCherry in Col-0 and *fer-4* with or without RALF23 treatment.
https://elifesciences.org/articles/74162/figures#fig1video2

*Herger et al., 2019*; *Herger et al., 2020*; *Moussu, 2020*). Their extensin domain confers cell wall anchoring, and their LRR domain mediates RALF binding (*Herger et al., 2019*; *Moussu, 2020*). Among the *Arabidopsis* 11-member *LRX* family, *LRX3*, *LRX4*, and *LRX5* are the most expressed in vegetative tissues, and the *lrx3 lrx4 lrx5* triple mutant (hereafter *lrx3/4/5*) shows stunted growth and salt hypersensitivity phenotypes reminiscent of *fer-4* (*Zhao et al., 2018*; *Dünser, 2019*). Therefore, we hypothesized that LRXs also regulate immune signaling. Indeed, co-immunoprecipitation experiments showed that *lrx3/4/5* was defective in flg22-induced FLS2-BAK1 complex formation (*Figure 2A*). Consistently, flg22-induced ROS production was reduced in *lrx3/4/5* similar to the levels observed in *fer-4* (*Figure 2B*). In addition, *lrx3/4/5* was impaired in elf18-induced ROS production (*Figure 2C*),

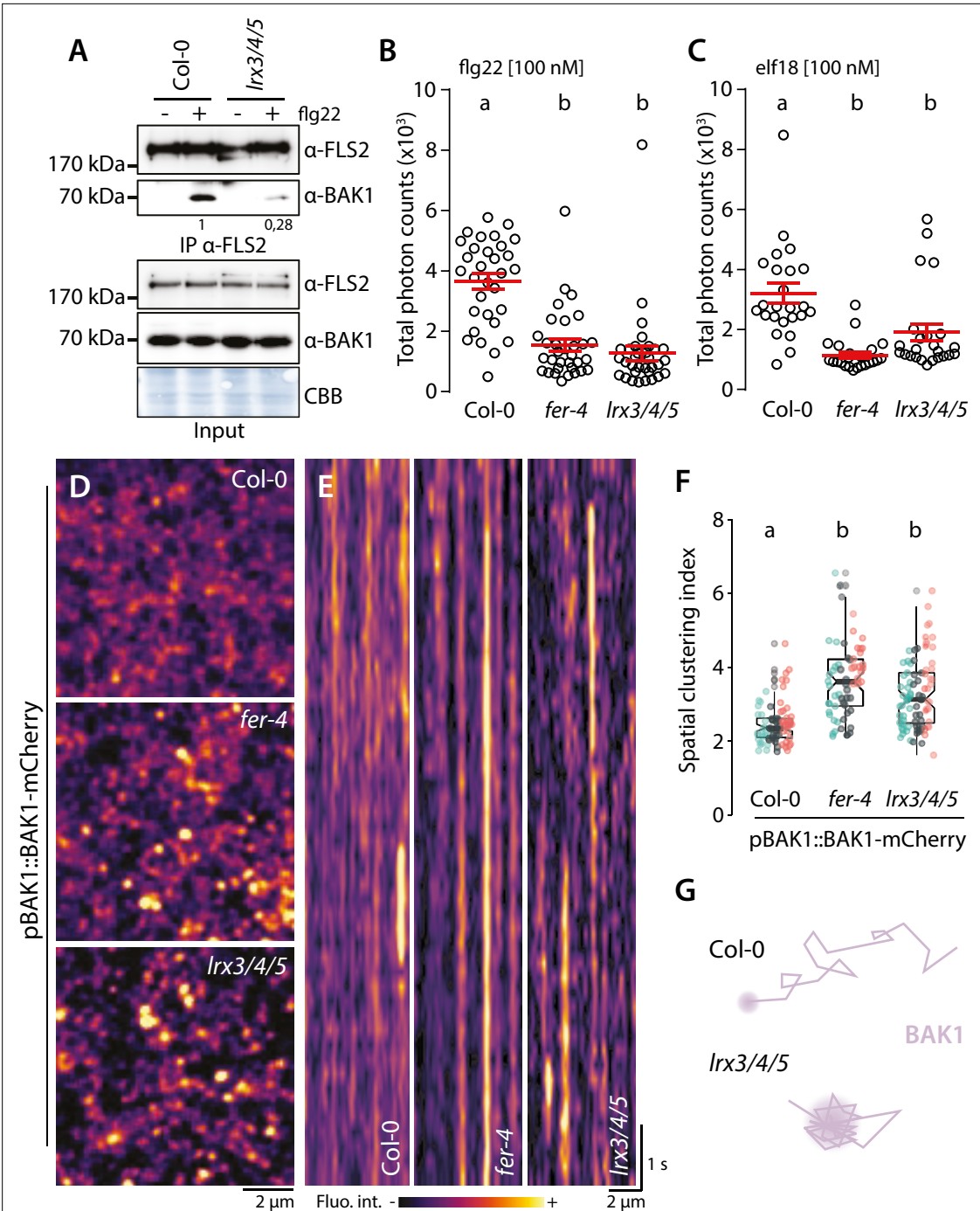

**Figure 2.** LRX3, LRX4, and LRX5 regulate pattern-triggered immunity (PTI) and BAK1-mCherry organization. (**A**) flg22-induced FLS2-BAK1 complex formation. Immunoprecipitation of FLS2 in *Arabidopsis* Col-0 and *lr3/4/5* seedlings either untreated or treated with 100 nM flg22 for 10 min. Blot stained with Coomassie brilliant blue (CBB) is presented to show equal loading. Western blots were probed with α-FLS2, α-BAK1, or α-FER antibodies. Numbers indicate quantification of BAK1 bands normalized based on the corresponding intensities of FLS2 bands and relative to the control Col-0. Similar results were obtained in at least three independent experiments. (**B, C**) Reactive oxygen species (ROS) production after elicitation with 100 nM elf18 (**B**) or 100 nM flg22 (**C**). Values are means of total photon counts over 40 min. Red crosses and red horizontal lines denote mean and SEM, n = 32. Conditions that do not share a letter are significantly different in Dunn's multiple comparison test (p<0.0001). (**D**) BAK1-mCherry nanodomain organization. Pictures are maximum projection images (10 variable angle total internal reflection fluorescence microscopy [VA-TIRFM] images obtained at 2.5 frames per second) of BAK1-mCherry in Col-0, *fer-4,* and *lrx3/4/5* cotyledon epidermal cells. (**E**) Representative kymograph showing lateral organization of BAK1-mCherry overtime in Col-0, *fer-4,* and *lrx3/4/5*. (**F**) Quantification of BAK1-mCherry spatial clustering index. Graphs are notched box plots, scattered data points show measurements, colors indicate independent experiments, n = 26 cells for Col-0/pBAK1::BAK1-mCherry, n = 31 cells for *fer-4/*pBAK1::BAK1-

*Figure 2 continued on next page*

*Figure 2 continued*

mCherry, n = 28 cells for *lrx3/4/5*/pBAK1::BAK1-mCherry. Conditions that do not share a letter are significantly different in Dunn's multiple comparison test (p<0.0001). (**G**) Graphical illustration summarizing our observations for BAK1-mCherry nanoscale dynamics in *lrx3/4/5*.

The online version of this article includes the following video, source data, and figure supplement(s) for figure 2:

**Source data 1.** Source data points for the graphs in *Figure 2B, C and F*.

**Source data 2.** Source blots images for the co-immunoprecipitation (co-IP) in *Figure 2A*.

**Figure supplement 1.** Subcellular localization of BAK1-mCherry in *fer-4* and *lrx3/4/5*.

**Figure supplement 1—source data 1.** Source data points for the graph in *Figure 2—figure supplement 1B*.

**Figure supplement 2.** *LRX3, LRX4,* and *LRX5* are dispensable for FER plasma membrane localization and accumulation.

**Figure supplement 2—source data 1.** Source blots images for *Figure 2—figure supplement 2*.

**Figure supplement 3.** *LRX3, LRX4,* and *LRX5* are dispensable for FER-GFP nanoscale organization.

**Figure supplement 3—source data 1.** Source data points for the graph in *Figure 2—figure supplement 3B*.

**Figure supplement 4.** LRX3, LRX4, and LRX5 contribute to RALF23 responsiveness.

**Figure supplement 4—source data 1.** Source data points for the graph in *Figure 2—figure supplement 4A and B*.

**Figure supplement 5.** RALF23 does not modulate constitutive association between FER and LRX4.

**Figure supplement 5—source data 1.** Source blots images for the co-immunoprecipitation (co-IP) in *Figure 2—figure supplement 5*.

**Figure 2—video 1.** Variable angle total internal reflection fluorescence microscopy (VA-TIRFM) imaging of BAK1-mCherry in Col-0 and *lrx3/4/5*.
https://elifesciences.org/articles/74162/figures#fig2video1

**Figure 2—video 2.** Variable angle total internal reflection fluorescence microscopy (VA-TIRFM) imaging of FER-GFP in *fer-4* and *fer-4;lrx3/4/5*.
https://elifesciences.org/articles/74162/figures#fig2video2

suggesting that, as for FLS2-BAK1 complex formation, *LRX3/4/5* are required for complex formation between EFR and BAK1. Thus, we conclude that LRX3/4/5 are positive regulators of PTI signaling.

We then asked whether, similar to FER, LRX3/4/5 regulate plasma membrane nanoscale organization. We crossed lines expressing FLS2-GFP and BAK1-mCherry under the control of their respective native promoter with the *lrx3/4/5* mutant. However, despite several attempts, we could not retrieve homozygous *lrx3/4/5* lines expressing FLS2-GFP. Nonetheless, VA-TIRFM and confocal imaging showed that, like in *fer-4*, BAK1-mCherry was more organized and more static in *lrx3/4/5* (*Figure 2D and E, Figure 2—video 1*), and that BAK1-mCherry plasma membrane localization was not affected by the loss of *LRX3/4/5* (*Figure 2—figure supplement 1*). Thus, like in *fer* mutants, perturbation in PTI signaling observed in *lrx3/4/5* correlates with alterations of plasma membrane RK organization.

LRX3, LRX4, and LRX5 have been proposed to sequester RALF peptides to prevent internalization of FER and inhibition of its function (*Zhao et al., 2018*). Following this logic, defects in PTI observed in *lrx3/4/5* could be explained by a depletion of FER at the plasma membrane. However, our confocal microscopy analysis and western blotting with anti-FER antibodies indicated that FER accumulation and plasma membrane localization were not affected in *lrx3/4/5* (*Figure 2—figure supplement 2*). Furthermore, VA-TIRFM revealed that FER-GFP transiently accumulated in dynamic foci, independently of *LRX3/4/5* (*Figure 2—figure supplement 3*, *Figure 2—video 2*). Together, these results suggest that LRX3/4/5 do not prevent RALF association with FER to modulate PTI. Moreover, our results suggest that active monitoring by the proposed cell wall integrity sensors FER and LRXs regulates plasma membrane nanoscale dynamics of RKs.

The ability of LRX3/4/5 to associate with RALF23 in planta (*Zhao et al., 2018*) prompted us to test whether LRX3/4/5 are required for RALF23 responsiveness. Indeed, *LRX3, LRX4,* and *LRX5* were required for RALF23-induced inhibition of elf18-triggered ROS production (*Figure 2—figure supplement 4A*). Similarly, we observed a decrease in RALF23-induced seedlings growth inhibition in *lrx3/4/5* compared to WT (*Figure 2—figure supplement 4B*). Altogether, these data show that LRX3/4/5 contribute to RALF23 responsiveness (*Figure 2—figure supplement 4C*), and that LRXs and FER have analogous functions in regulating PTI.

We next asked whether FER and LRX3/4/5 form a complex. For this, we made use of a truncated version of LRX4 lacking its extensin domain (LRX4$^{\Delta E}$), previously used to assess protein complex formation (*Dünser, 2019*; *Herger et al., 2020*). Consistent with previous reports based on transient expression in *Nicotiana benthamiana* (*Dünser, 2019*; *Herger et al., 2020*), co-immunoprecipitation

experiments with stable transgenic *Arabidopsis* showed that FER was constitutively associated with LRX4$^{\Delta E}$-FLAG, and that RALF23 treatment did not modulate this interaction (*Figure 2—figure supplement 5*). This suggests that direct monitoring of the cell wall mediated by a possible FER-LRX complex (*Dünser, 2019*; *Herger et al., 2019*) is not regulated by RALF23. In agreement with structural and biochemical analyses of RALF-binding by CrRLK1Ls/LLGs and LRXs (*Moussu, 2020*), FER-LLG1 and LRX3/4/5 may form distinct RALF23 receptor complexes. Similar to their roles in pollen tube and root hair growth and integrity (*Ge et al., 2017*; *Mecchia, 2017*; *Moussu, 2020*; *Dünser, 2019*; *Herger et al., 2020*), future investigations are thus needed to understand the exact molecular link between RALF-binding LRXs and CrRLK1s.

## Functional dichotomy of FER and LRXs in regulating growth and immunity

In line with previous reports, our data show that FER and LRXs can form a complex (*Dünser, 2019*; *Herger et al., 2019*, *Figure 2—figure supplement 5*). Moreover, they are known to associate with the cell wall (*Baumberger et al., 2001*; *Feng, 2018*) and are proposed to cooperatively relay its properties (*Dünser, 2019*; *Herger et al., 2019*). We thus asked if direct cell wall sensing underlies FER and LRXs function in PTI. In the context of growth and cell expansion, plants overexpressing LRX4$^{\Delta E}$ are phenotypically reminiscent of *lrx3/4/5* and *fer-4* mutants (*Dünser, 2019*). This dominant negative effect is proposed to be caused by competition of the overexpressed truncated LRX4$^{\Delta E}$ with endogenous LRXs and consequent loss of cell wall anchoring (*Dünser, 2019*). Similarly, overexpression of LRX1$^{\Delta E}$ inhibits root hair elongation, phenocopying *LRX1/LRX2* loss of function (*Herger et al., 2020*). By contrast, we observed that LRX4$^{\Delta E}$ overexpression did not affect flg22-induced interaction between FLS2 and BAK1 (*Figure 3—figure supplement 1A*). In good agreement with this notion, overexpression of LRX4$^{\Delta E}$ did not affect flg22- nor elf18-induced ROS production (*Figure 3—figure supplement 1B and C*). To corroborate these results, we tested inhibition of root growth triggered by flg22 treatment. Consistent with the positive role of FER and LRX3/4/5 in PTI, we observed that *fer-4* and *lrx3/4/5* were hyposensitive to flg22 treatment (*Figure 3—figure supplement 1D*). By contrast, overexpression of LRX4$^{\Delta E}$ did not affect inhibition of root growth by flg22 (*Figure 3—figure supplement 1D*). In addition, we observed that LRX4$^{\Delta E}$ overexpression did not impact RALF23 responsiveness (*Figure 3—figure supplement 1E*). Altogether, these data suggest that the function of LRX3/4/5 in PTI is distinct from their role during growth.

The ectodomain of FER contains two malectin-like domains, malA and malB (*Figure 3A*), which share homology with malectin, a carbohydrate-binding protein from *Xenopus laevis* (*Boisson-Dernier et al., 2011*). Despite lacking the canonical carbohydrate-binding site of malectin (*Moussu, 2018*; *Xiao et al., 2019*), malA and malB were proposed to bind pectin (*Feng, 2018*; *Lin et al., 2021*), and FER-mediated cell wall sensing regulates pavement cell and root hair morphogenesis (*Duan, 2010*; *Lin et al., 2021*). To investigate if direct cell wall sensing underlies FER's function in regulating PTI, we used transgenic lines expressing a FER truncated mutant, lacking the malA domain, C-terminally fused to YFP (FER$^{\Delta malA}$-YFP) in the *fer-4* mutant background (*Figure 3B*). We observed that FER$^{\Delta malA}$-YFP did not complement the cell shape and root hair elongation defects of *fer-4* (*Figure 3C and D*), emphasizing the importance of malA in FER-regulated cell morphogenesis. In contrast, immunoprecipitation assays showed that FER$^{\Delta malA}$-YFP fully complemented flg22-induced complex formation between endogenous FLS2 and BAK1 (*Figure 3E*) as well as ROS production in response to flg22 and elf18 (*Figure 3F and G*). Altogether, these data suggest that malA-mediated cell wall sensing underlies specific function(s) of FER in regulating growth and cell morphology, but is dispensable for FER's role in PTI. Interestingly, we observed that expression of FER$^{\Delta malA}$-YFP restored inhibition of growth triggered by RALF23, suggesting that malB is sufficient for RALF responsiveness (*Figure 3H*), as suggested by its physical interaction with RALF23 (*Xiao et al., 2019*). While we cannot formally exclude the implication of pectin-binding by malB in regulating immunity, the contrasted context-dependent functionality of FER$^{\Delta malA}$-YFP suggests that FER's function in PTI is primarily mediated by RALF perception. Altogether, our data indicate molecular and functional dichotomy of FER and LRXs in regulating growth and immunity.

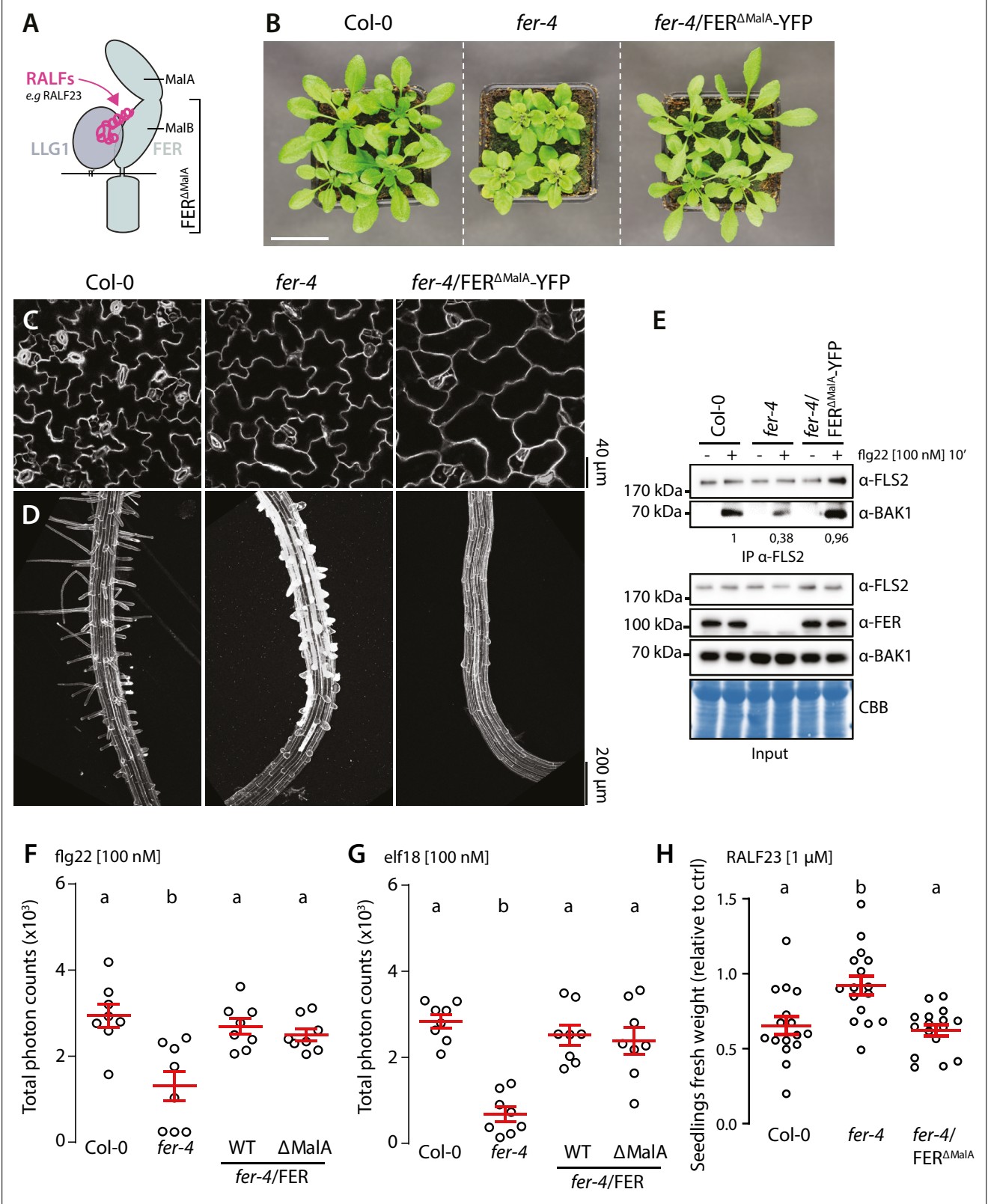

**Figure 3.** FER malectin A domain regulates cell morphogenesis not pattern-triggered immunity (PTI). (**A**) Graphical representation of RALF23 perception by FER-LLG1 complex. (**B**) Morphology of 4-week-old *Arabidopsis* plants; scale bar indicates 5 cm. (**C, D**) Confocal microscopy pictures of 5-day-old seedlings cotyledon (**C**) and root (**D**) epidermal cells stained with propidium iodide. 3–4 seedlings per genotypes were observed per experiment. For each seedling, we observed the center of both cotyledons, and at the initiation site of root hairs. Similar results were obtained in at

*Figure 3 continued on next page*

*Figure 3 continued*

least three independent experiments. (**E**) Flg22-induced FLS2-BAK1 complex formation. Immunoprecipitation of FLS2 in *Arabidopsis* Col-0, *fer-4*, and *fer-4*/p35S::FER^ΔMalA-YFP seedlings that were either untreated or treated with 100 nM flg22 for 10 min. Blot stained with Coomassie brilliant blue (CBB) is presented to show equal loading. Western blots were probed with α-FLS2, α-BAK1, or α-FER antibodies. Numbers indicate quantification of BAK1 bands normalized based on the corresponding intensities of FLS2 bands and relative to the control Col-0 + flg22. Similar results were obtained in at least three independent experiments. (**F, G**) Reactive oxygen species (ROS) production after elicitation with 100 nM flg22 (**F**) or 100 nM elf18 (**G**). Values are means of total photon counts over 40 min, n = 8. Red crosses and red horizontal lines denote mean and SEM, respectively. Conditions that do not share a letter are significantly different in Dunn's multiple comparison test (p<0.0001). (**H**) Fresh weight of 12-day-old seedlings grown in the absence (mock) or presence of 1 µM of RALF23 peptide. Fresh weight is expressed as relative to the control mock treatment for each genotype. Similar results were obtained in at least three independent experiments. Conditions that do not share a letter are significantly different in Dunn's multiple comparison test (p<0.001).

The online version of this article includes the following source data and figure supplement(s) for figure 3:

**Source data 1.** Source data points for the graphs in *Figure 3F–H*.

**Source data 2.** Source blots images for the co-immunoprecipitation (co-IP) in *Figure 3E*.

**Figure supplement 1.** Overexpression of LRX4^ΔE does not affect pattern-triggered immunity (PTI).

**Figure supplement 1—source data 1.** Source data points for the graphs in *Figure 3—figure supplement 1B–E*.

**Figure supplement 1—source data 2.** Source blots images for the co-immunoprecipitation (co-IP) in *Figure 3—figure supplement 1A*.

## RALF23 alters FLS2 and BAK1 organization and function through active FER signaling

We next asked whether RALF23 activity is mediated by active FER signaling. We used a kinase-dead mutant (FER^K565R) C-terminally fused to GFP, expressed in *fer* knock-out backgrounds (*Chakravorty et al., 2018*), and selected lines showing comparable accumulation to endogenous FER in WT (*Figure 4—figure supplements 1 and 2*). Interestingly, we observed that FER^K565R-GFP complemented *fer*'s defect in FLS2-BAK1 complex formation (*Figure 4—figure supplement 1A*) and PAMP-induced ROS production (*Figure 4—figure supplement 1B and C*). In contrast, we observed that inhibition of FLS2-BAK1 complex formation by RALF23 depended on FER kinase activity (*Figure 4—figure supplement 2B*). Similarly, inhibition of elf18-induced ROS production and seedling growth inhibition by RALF23 depended on FER kinase activity (*Figure 4—figure supplement 2C*). Overall, these data show that inhibition by RALF23 is mediated by active FER signaling while FER's positive role in immune signaling is kinase activity-independent.

We next asked whether inhibition of FLS2-BAK1 complex formation by RALF23 correlates with a modulation of FLS2 or BAK1 nanoscale organization. VA-TIRFM imaging showed an increase of FLS2-GFP mobility and an alteration of FLS2-GFP nanodomain organization within minutes of RALF23 treatment (*Figure 4—figure supplements 3 and 4*, *Figure 4—videos 1–3*, imaging performed 2–30 min post treatment; *Figure 4—figure supplement 5*). In addition, we observed that RALF23 treatment stabilized BAK1-mCherry nanoscale organization (*Figure 4*, *Figure 4—figure supplement 6*, *Figure 1—video 2*). These data suggest that RALF23 perception leads to rapid modification of FLS2 and BAK1 membrane organization and thereby potentially inhibits their association. In addition, these data, based on short-term RALF23 treatment, demonstrate that the aforementioned defects in FLS2 and BAK1 organization observed in *fer* and *lrx3/4/5* mutant plants are not caused by their pleiotropic growth defects.

Our study unravels the regulation of FLS2 and BAK1 nanoscale organization by the RALF receptors FER and LRX3/4/5 (*Figure 4—figure supplement 6*). The function of RALF receptors in other processes might similarly rely on the regulation of RK nanoscale dynamics, and the identification of the corresponding regulated RKs is an exciting prospect for future investigation. Further work will be required to decipher how FLS2 and BAK1 associate in a ligand-dependent manner within the plasma membrane and to understand how FER and LRXs control this process. While FER associate with LLG1 to perceive RALF peptides, whether perception of these peptides by LRXs involves additional unknown components remains open. For both FER-LLG1 and LRXs, it will be important in the future to identify the components mediating RALF23 signaling and modification of FLS2 and BAK1 nanoscale dynamics. Because FER-LLG1 and LRX3/4/5 – components of distinct RALFs receptor complexes – are genetically required to control FLS2 and BAK1 nanoscale dynamics, we hypothesize that perception of additional RALF peptides may be involved in regulating this process (*Figure 4—figure supplement*

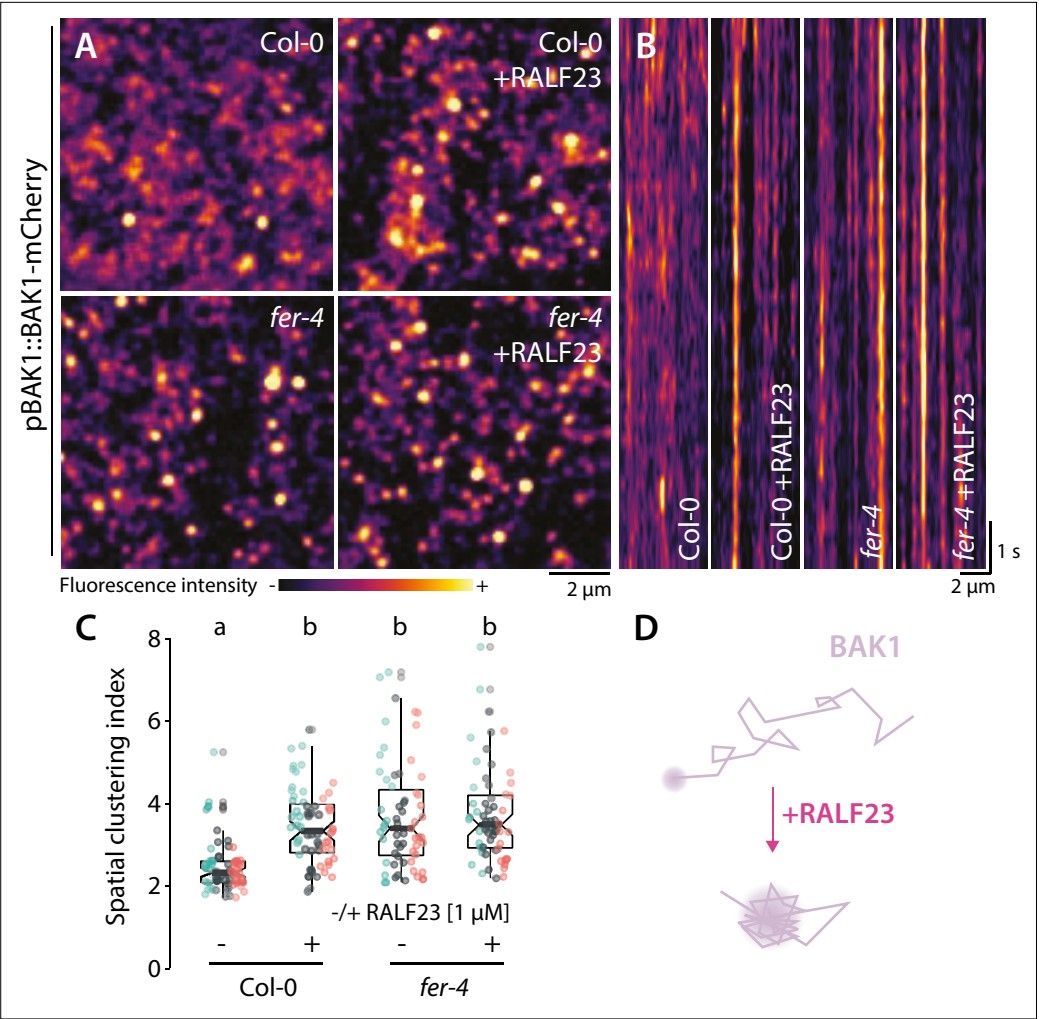

**Figure 4.** RALF23 perception regulates BAK1-mCherry organization. (**A**) BAK1-mCherry nanodomain organization (pBAK1::BAK1-mCherry). Pictures are maximum projection images (10 variable angle total internal reflection fluorescence microscopy [VA-TIRFM] images obtained at 2.5 frames per second) of BAK1-mCherry in Col-0 and *fer-4* cotyledon epidermal cells with or without 1 μM RALF23 treatment (2–30 min). (**B**) Representative kymograph showing lateral organization of BAK1-mCherry overtime in Col-0 and *fer-4* with or without 1 μM RALF23 treatment. (**C**) Quantification of BAK1-mCherry spatial clustering index. Graphs are notched box plots, scattered data points show measurements, colors indicate independent experiments, n = 21 and n = 23 cells for Col-0/pBAK1::BAK1-mCherry with and without RALF23, respectively, n = 20 and n = 21 cells for *fer-4*/pBAK1::BAK1-mCherry with and without RALF23, respectively. Conditions that do not share a letter are significantly different in Dunn's multiple comparison test (p<0.0001). (**D**) Graphical illustration summarizing our observations for BAK1-mCherry nanoscale dynamics upon RALF23 treatment.

The online version of this article includes the following video, source data, and figure supplement(s) for figure 4:

**Source data 1.** Source data points for the graph in *Figure 4C*.

**Figure supplement 1.** FER kinase activity is dispensable to support pattern-triggered immunity (PTI) signaling.

**Figure supplement 1—source data 1.** Source data points for the graphs in *Figure 4—figure supplement 1B and C*.

**Figure supplement 2.** Inhibition of pattern-triggered immunity (PTI) signaling by RALF23 requires FER kinase activity.

**Figure supplement 2—source data 1.** Source data points for the graph in *Figure 4—figure supplement 2C*.

**Figure supplement 3.** Analysis of FLS2-GFP single-particle dynamics upon RALF23 treatment.

**Figure supplement 3—source data 1.** Source data points for the graph in *Figure 4—figure supplement 3B*.

**Figure supplement 4.** Analysis of FLS2-GFP organization upon RALF23 treatment.

*Figure 4 continued on next page*

*Figure 4 continued*

**Figure supplement 4—source data 1.** Source data points for the graph in *Figure 4—figure supplement 4C*.

**Figure supplement 5.** Time-resolved analysis of the spatial clustering index.

**Figure supplement 5—source data 1.** Source data points for the graphs in *Figure 4—figure supplement 5A and B*.

**Figure supplement 6.** Working model for the regulation of FLS2 and BAK1 nanoscale organization by RALFs receptor complexes.

**Figure supplement 7.** Linear regression analysis of the relationship between the spatial clustering index and fluorescence intensity.

**Figure supplement 7—source data 1.** Source data points for the graphs in *Figure 4—figure supplement 6A–C*.

**Figure supplement 8.** Replicates of co-immunoprecipitation experiments.

**Figure supplement 8—source data 1.** Source blots images for the replicate of co-immunoprecipitation (co-IP) experiments.

**Figure 4—video 1.** Variable angle total internal reflection fluorescence microscopy (VA-TIRFM) imaging of FLS2-GFP in Col-0 with or without RALF23 treatment.

https://elifesciences.org/articles/74162/figures#fig4video1

**Figure 4—video 2.** Variable angle total internal reflection fluorescence microscopy (VA-TIRFM) imaging of FLS2-GFP in *fer-4* with or without RALF23 treatment.

https://elifesciences.org/articles/74162/figures#fig4video2

**Figure 4—video 3.** Variable angle total internal reflection fluorescence microscopy (VA-TIRFM) imaging of FLS2-GFP in Col-0 and *fer-2* with or without RALF23 treatment.

https://elifesciences.org/articles/74162/figures#fig4video3

*6*). Plants have evolved coordinated RK protein-protein interaction networks to process extracellular signals into specific responses (*Smakowska-Luzan et al., 2018*), and thus may have co-evolved mechanisms to regulate these interactions in both space and time. Our results suggest that perception of endogenous peptides by distinct receptor complexes actively modulates the plasma membrane nanoscale organization to regulate cell surface signaling by other RKs.

## Materials and methods

**Key resources table**

| Reagent type (species) or resource | Designation | Source or reference | Identifiers | Additional information |
|---|---|---|---|---|
| Genetic reagent (*Arabidopsis thaliana*) | Col-0/pFLS2::FLS2-GFP#1 | *Göhre et al., 2008* | | See Materials and methods |
| Genetic reagent (*A. thaliana*) | Col-0/pFLS2::FLS2-GFP#2 | This paper | | See Materials and methods |
| Genetic reagent (*A. thaliana*) | *fer-2*/pFLS2::FLS2-GFP#1 | *Stegmann et al., 2017* | | See Materials and methods |
| Genetic reagent (*A. thaliana*) | *fer-4*/pFLS2::FLS2-GFP#2 | This paper | | See Materials and methods |
| Genetic reagent (*A. thaliana*) | *fer-4* | *Duan, 2010* | | See Materials and methods |
| Genetic reagent (*A. thaliana*) | *fer-4*/pFER::FER-GFP | *Duan, 2010* | | See Materials and methods |
| Genetic reagent (*A. thaliana*) | *fer-4*/pFER::FERKD-GFP | *Chakravorty et al., 2018* | | See Materials and methods |
| Genetic reagent (*A. thaliana*) | *lrx3/4/5* | *Dünser, 2019* | | See Materials and methods |
| Genetic reagent (*A. thaliana*) | p35S::LRX4ΔE-Citrine | *Dünser, 2019* | | See Materials and methods |
| Genetic reagent (*A. thaliana*) | p35S::LRX4ΔE-FLAG | *Dünser, 2019* | | See Materials and methods |
| Genetic reagent (*A. thaliana*) | Col-0/pBAK1::BAK1-mCherry | *Bücherl et al., 2013* | | See Materials and methods |
| Genetic reagent (*A. thaliana*) | *fer-4*/ pBAK1::BAK1-mCherry | This paper | | See Materials and methods |

*Continued on next page*

*Continued*

| Reagent type (species) or resource | Designation | Source or reference | Identifiers | Additional information |
|---|---|---|---|---|
| Genetic reagent (*A. thaliana*) | *lrx3/4/5/* pBAK1::BAK1-mCherry | This paper | | See Materials and methods |
| Genetic reagent (*A. thaliana*) | *lrx3/4/5/* pFER::FER-GFP | This paper | | See Materials and methods |
| Antibody | anti-FLAG-HRP | Sigma-Aldrich | A8592 | WB (1:4000 dilution) |
| Antibody | Monoclonal rabbit anti-FLS2 | *Chinchilla et al., 2007* | | WB (1:1000 dilution) |
| Antibody | Polyclonal rabbit anti-BAK1 | *Roux, 2011* | | WB (1:5000 dilution) |
| Antibody | Polyclonal rabbit anti-BAK1 pS612 | *Perraki, 2018* | | WB (1:3000 dilution) |
| Antibody | Polyclonal rabbit anti-FER | *Xiao et al., 2019* | | WB (1:2000 dilution) |
| Antibody | Anti-rabbit IgG-HRP Trueblot | Rockland | 18-8816-31 | WB (1:10,000 dilution) |
| Peptide, recombinant protein | Flg22 | Synthesized by EZBiolab (purity >95%) | | See Materials and methods |
| Peptide, recombinant protein | Elf18 | Synthesized by EZBiolab (purity >95%) | | See Materials and methods |
| Peptide, recombinant protein | RALF23 | Synthesized by EZBiolab (purity >95%) | | See Materials and methods |
| Chemical compound, drug | GFP-Trap agarose beads | ChromoTek | | See Materials and methods |
| Chemical compound, drug | M2 anti-Flag affinity gel | Sigma-Aldrich | A2220-5ML | See Materials and methods |
| Chemical compound, drug | Anti-rabbit Trueblot agarose beads | eBioscience | SML1656 | See Materials and methods |
| Software, algorithm | Fiji | https://imagej.net/Fiji | | See Materials and methods |

## Plant materials and growth

*A. thaliana* ecotype Columbia (Col-0) was used as WT control. The *fer-4, fer-4*/pFER::FER-GFP (*Duan, 2010*), *fer-4*/pFER::FER$^{KD}$-GFP (*Chakravorty et al., 2018*), *fer-4*/p35S::FER$^{ΔMalA}$-GFP (*Lin, 2018*), Col-0/pFLS2::FLS2-GFP#1 (*Göhre et al., 2008*), Col-0/pFLS2::FLS2-GFP#2 (this study), *fer-2*/pFLS2::FLS2-GFP (*Stegmann et al., 2017*), *lrx3/4/5*, p35S::LRX4$^{ΔE}$-Citrine and p35S::LRX4$^{ΔE}$-FLAG (*Dünser, 2019*) lines were previously published. Col-0/pFLS2::FLS2-GFP (*Göhre et al., 2008*) was crossed with *fer-4* to obtain *fer-4*/pFLS2::FLS2-GFP. Col-0/pBAK1::BAK1-mCherry (*Bücherl et al., 2013*) was crossed with *fer-4* and *lrx3/4/5* to obtain *fer-4*/pBAK1::BAK1-mCherry and *lrx3/4/5*/pBAK1::BAK1-mCherry. *fer-4*/pFER::FER-GFP was crossed with *lrx3/4/5* to obtain *fer-4*/*lrx3/4/5*;pFER::FER-GFP. For the VA-TIRFM imaging, we initially used a line expressing pFLS2::FLS2-GFP in *fer-2* we previously generated (*Stegmann et al., 2017*). Despite that both alleles are well-characterized null allele of *FER*, for consistent and direct comparison of our biochemical, physiological, and imaging experiments, we also crossed another Col-0/pFLS2::FLS2-GFP with *fer-4*. To further facilitate the single-particle tracking analysis, we choose a Col-0/pFLS2::FLS2-GFP line expressing less FLS2-GFP. For ROS burst assays, plants were grown in individual pots at 20–21°C with a 10 hr photoperiod in environmentally controlled growth rooms. For seedling-based assays, seeds were surface-sterilized using chlorine gas for 5 hr and grown at 22°C and a 16 hr photoperiod on Murashige and Skoog (MS) medium supplemented with vitamins, 1% sucrose and 0.8% agar.

## Synthetic peptides and chemicals

The flg22 (QRLSTGSRINSAKDDAAGLQIA), elf18 (SKEKFERTKPHVNVGTIG), and RALF23 (ATTKYISY GALRRNTVPCSRRGASYYNCRRGAQANPYSRGCSAITRCRR) peptides were synthesized by EZBiolab (USA) with a purity of >95%. All peptides were dissolved in sterile purified water.

## ROS burst measurement

ROS burst measurements were performed as previously documented (*Kadota et al., 2014*). At least eight leaf discs (4 mm in diameter) per individual genotype were collected in 96-well plates containing sterile water and incubated overnight. The next day the water was replaced by a solution containing 17 µg/mL luminol (Sigma-Aldrich), 20 µg/mL horseradish peroxidase (HRP, Sigma-Aldrich), and the peptides in the appropriate concentration. Luminescence was measured for the indicated time period

using a charge-coupled device camera (Photek Ltd., East Sussex, UK). The effect of RALF23 on elf18-triggered ROS production was performed as previously described (*Stegmann et al., 2017*). 8–10 leaf discs per treatment and/or genotype were collected in 96-well plates containing water and incubated overnight. The following day the water was replaced by 75 µL of 2 mM MES-KOH pH 5.8 to mimic the apoplastic pH. Leaf discs were incubated further for 4–5 hr before adding 75 µL of a solution containing 40 µg/mL HRP, 1 µM L-O12 (Wako Chemicals, Germany), and 2× elicitor RALF peptide solution (final concentration 20 µg/mL HRP, 0.5 µM L-O12, 1× elicitors). ROS production is displayed as the integration of total photon counts.

## Root growth inhibition assay

Three-day-old Col-0, *fer-4*, *lrx3/4/5*, and 35S::LRR4-Cit seedlings (n = 9–12) were transferred for additional 3 days to 3 mL liquid ½ MS medium containing different concentrations (100 nM, 300 nM, or 1 µM) of flg22 or the appropriate amount of solvent. The seedlings were then placed on solid MS plates before scanning. Root length was measured using ImageJ.

## Live-cell imaging

For confocal microscopy and VA-TIRF microscopy experiments, surface-sterilized seeds were individually placed in line on square Petri dishes containing 1/2 MS 1% sucrose, 0.8% phytoagar, stratified 2 days in the dark at 4°C, then placed in a growth chamber at 22°C and a 16 hr photoperiod for 5 days. Seedlings were mounted between a glass slide and a coverslip in liquid 1/2 MS, 1% sucrose medium. For VA-TIRF microscopy experiments, 2–4 seedlings were sequentially imaged for each genotype and/or condition. To test the effect of RALF23 on FLS2-GFP dynamics and nanodomain organization, seedlings were preincubated in 2 mM MES-KOH pH 5.8 for 3–4 hr prior treatment. Seedlings were imaged 2–30 min after treatment.

## Confocal laser scanning microscopy (CLSM)

Confocal microscopy was performed using a Leica SP5 CLSM system (Leica, Wetzlar, Germany) equipped with Argon, DPSS, He-Ne lasers, hybrid detectors, and using a 63 × 1.2 NA oil immersion objective. GFP was excited using 488 nm argon laser, and emission wavelengths were collected between 495 and 550 nm. mCherry was excited using 561 nm He/Ne laser, and emission wavelengths were collected between 570 and 640 nm. Propidium iodide was imaged using 488 nm and 500–550 nm excitation and emission wavelengths, respectively. In order to obtain quantitative data, experiments were performed using strictly identical confocal acquisition parameters (e.g., laser power, gain, zoom factor, resolution, and emission wavelengths reception), with detector settings optimized for low background and no pixel saturation. Pseudo-color images were obtained using look-up-table (LUT) of Fiji software (*Schindelin et al., 2012*).

## Total internal reflection fluorescence (TIRF) microscopy

TIRF microscopy was performed using an inverted Leica GSD equipped with a ×160 objective (NA = 1.43, oil immersion), and an Andor iXon Ultra 897 EMCCD camera. Images were acquired by illuminating samples with a 488 nm solid-state diode laser set at 15 mW using a cube filter with an excitation filter 488/10 and an emission filter 535/50 for FLS2-GFP and FER-GFP. Optimum critical angle was determined as giving the best signal-to-noise for our sample and was kept fixed for each experiment. Images time series were recorded at 20 frames per second (50 ms exposure time) for *Figure 1—figure supplement 2* and *Figure 4—figure supplements 3 and 4*; 5 frames per second for *Figure 1A–C* and *Figure 2—figure supplement 3*. To observe BAK1-mCherry, we could only use a 532 nm solid-state diode laser (ca. 40% of maximum excitation for mCherry) using a cube filter with an excitation filter 532/10 and an emission filter 600/100. To obtain a sufficient signal-to-noise ratio, image time series were recorded at 2.5 frames per second (*Figures 1, 2 and 4*). Due to apparent high mobility of BAK1 and relatively slow acquisition rate, we could not asses with confidence the identity of fluorescent particles from one time frame to another and therefore did not perform particle tracking analysis of BAK1-mCherry. VA-TIRFM images were subjected to background subtraction (30 rolling pixel radius) and smoothing. Kymographs were generated using Orthogonal views in Fiji (*Schindelin et al., 2012*).

## Single-particle tracking analysis

To analyze single-particle tracking experiments, we used the plugin TrackMate 2.7.4 (*Tinevez et al., 2017*) in Fiji (*Schindelin et al., 2012*). Single particles were segmented frame-by-frame by applying

a Laplacian of Gaussian (LoG) filter and estimated particle size of 0.4 μm. Individual single particles were localized with sub-pixel resolution using a built-in quadratic fitting scheme. Then, single-particle trajectories were reconstructed using a simple linear assignment problem (*Jaqaman et al., 2008*) with a maximal linking distance of 0.4 μm and without gap closing. Thresholds were empirically determined for optimal single-particle detection and linking. Only tracks with at least 10 successive points (tracked for 500 ms) were selected for further analysis. Diffusion coefficients of individual particles were determined using TraJClassifier (*Wagner et al., 2017*). For each particle, the slope of the first four time points of their mean square displacement (MSD) plot was used to calculate their diffusion coefficient according to the following equation: $MSD = (x - x_0)^2 + (y - y_0)^2$ and $D = MSD/4t$, where x0 and y0 are the initial coordinates, and x and y are the coordinates at any given time, and *t* is the time frame.

## Quantification of SCI

Genotype and/or treatment-dependent variation in fluorescence intensity of FLS2-GFP and fluorescence pattern of FLS2-GFP and BAK1-mCherry compromised the use of a unique set of parameters to compute nanodomain size and density across the different experiments. To uniformly quantify differences in membrane organization of both FLS2 and BAK1 across all experiments, we used the SCI that was shown to be largely insensitive to variation in fluorescence intensity (*Gronnier et al., 2017*). Quantifications were performed as previously described (*Gronnier et al., 2017*). Briefly, fluorescence intensity was plotted along an 8-μm-long line on maximum projection of VA-TIRFM images. Three plots were randomly recorded per cell and at least eight cells per condition per experiment were analyzed. For each line plot, the SCI was calculated by dividing the mean of the 5% highest values by the mean of 5% lowest values. Because the absence of correlation between fluorescence intensity and SCI was assessed on confocal microscopy images and for a single protein (*Gronnier et al., 2017*), we tested whether this was also the case in our experimental conditions. Indeed, we consistently observed poor to no correlation between variation in fluorescence intensity and values of SCI (*Figure 4—figure supplement 7*).

## Co-immunoprecipitation experiments

20–30 seedlings per plate were grown in wells of a 6-well plate for 2 weeks, transferred to 2 mM MES-KOH, pH 5.8, and incubated overnight. The next day flg22 (final concentration 100 nM) and/or RALF23 (final concentration 1 μM) were added and incubated for 10 min. Seedlings were then frozen in liquid N2 and subjected to protein extraction. To analyze FLS2-BAK1 receptor complex formation, proteins were isolated in 50 mM Tris-HCl pH 7.5, 150 mM NaCl, 10% glycerol, 5 mM dithiothreitol, 1% protease inhibitor cocktail (Sigma-Aldrich), 2 mM $Na_2MoO_4$, 2.5 mM NaF, 1.5 mM activated $Na_3VO_4$, 1 mM phenylmethanesulfonyl fluoride, and 0.5% IGEPAL. For immunoprecipitations, α-rabbit True-blot agarose beads (eBioscience) coupled with α-FLS2 antibodies (*Chinchilla et al., 2007*) or GFP-Trap agarose beads (ChromoTek) were used and incubated with the crude extract for 3–4 hr at 4°C. Subsequently, beads were washed three times with wash buffer (50 mM Tris-HCl pH 7.5, 150 mM NaCl, 1 mM phenylmethanesulfonyl fluoride, 0,1% IGEPAL) before adding Laemmli sample buffer and incubating for 10 min at 95°C. Analysis was carried out by SDS-PAGE and immunoblotting. To test the association between Flag-LRX4 and FER, total protein from 60 to 90 seedlings per treatment per genotype was extracted as previously described. For immunoprecipitations, M2 anti-Flag affinity gel (Sigma A2220-5ML) was used and incubated with the crude extract for 2–3 hr at 4°C. Subsequently, beads were washed three times with wash buffer (50 mM Tris-HCl pH 7.5, 150 mM NaCl, 1 mM phenylmethanesulfonyl fluoride, 0.1% IGEPAL) before adding Laemmli sample buffer and incubating for 10 min at 95°C. Analysis was carried out by SDS-PAGE and immunoblotting. The replicates of the co-immunoprecipitation are presented in *Figure 4—figure supplement 8*.

## Immunoblotting

Protein samples were separated in 10% bisacrylamide gels at 150 V for approximately 2 hr and transferred into activated PVDF membranes at 100 V for 90 min. Immunoblotting was performed with antibodies diluted in blocking solution (5% fat-free milk in TBS with 0.1% [v/v] Tween-20). Antibodies used in this study were α-BAK1 (1:5000; *Roux, 2011*), α-FLS2 (1:1000; *Chinchilla et al., 2007*), α-FER (1:2000; *Xiao et al., 2019*), α-BAK1 pS612 (1:3000; *Perraki, 2018*), α-FLAG-HRP (Sigma-Aldrich, A8592, dilution 1:4000), and α -GFP (sc-9996, Santa Cruz, used at 1:5000). Blots were developed with

Pierce ECL/ECL Femto Western Blotting Substrate (Thermo Scientific). The following secondary antibodies were used: anti-rabbit IgG-HRP Trueblot (Rockland, 18-8816-31, dilution 1:10,000) for detection of FLS2-BAK1 co-immunoprecipitation or anti-rabbit IgG (whole molecule)–HRP (A0545, Sigma, dilution 1:10,000) for all other western blots.

## Statistical analysis

Statistical analyses were carried out using Prism 6.0 software (GraphPad). As mentioned in the figure legends, statistical significances were assessed using nonparametric Kruskal–Wallis bilateral tests combined with post-hoc Dunn's multiple pairwise comparisons or using a two-way nonparametric Student's $t$-test Mann–Whitney test.

## Accession numbers

FER (AT3G51550), LRX3 (AT4G13340), LRX4 (AT3G24480), LRX5 (AT4G18670), RALF23 (AT3G16570), FLS2 (AT5G46330), BAK1 (AT4G33430).

## Acknowledgements

We thank all present and past members of the Zipfel laboratory for fruitful discussions and comments on the manuscript. We thank the members of the Grossniklaus, Ringli, Sanchez-Rodriguez, and Keller laboratories for sharing results and comments during our stimulating CCWI meetings. We thank Vera Gorelova, Yvon Jaillais, Alexandre Martinière, and Birgit Kemmerling for comments on the manuscript. This research was funded by the Gatsby Charitable Foundation (CZ), the University of Zürich (CZ), the European Research Council under the Grant Agreements 309858 and 773153 (grants PHOS-PHinnATE and IMMUNO-PEPTALK to CZ) and 639678 (grant AuxinER to JK-V), the Swiss National Science Foundation (grant no. 31003A_182625 to CZ and 31003A_166577/1 to CR), and the Austrian science fund (FWF; P33044 to JK-V). JG, CMF, and TAD were supported by Long-Term Fellowships from the European Molecular Biology Organization (EMBO) (numbers 438-2018, 512-2019, and 100-2017, respectively), while MS was supported by a postdoctoral fellowship (STE 2448/1) from the Deutsche Forschungsgemeinschaft (DFG) and KD by a doctoral fellowship from the Austrian Academy of Sciences (ÖAW). We thank Sarah Assman, Sacco de Vries, Silke Robatzek, and Nana Keinath for kindly providing segregating lines of fer-4/pFER::FER$^{K565R}$-GFP, Col-0/pBAK1::BAK1-mCherry, Col-0/pFLS2::FLS2-GFP, and fer-2/pFER::FER$^{K565R}$-GFP, respectively.

## Additional information

### Competing interests

Jürgen Kleine-Vehn: Senior editor, eLife. The other authors declare that no competing interests exist.

### Funding

| Funder | Grant reference number | Author |
|---|---|---|
| Gatsby Charitable Foundation | | Cyril Zipfel |
| University of Zurich | | Cyril Zipfel |
| H2020 European Research Council | 309858 | Cyril Zipfel |
| H2020 European Research Council | 773153 | Cyril Zipfel |
| European Molecular Biology Organization | LTF 438-2018 | Julien Gronnier |
| European Molecular Biology Organization | LTF 512-2019 | Christina M Franck |

| Funder | Grant reference number | Author |
|---|---|---|
| European Molecular Biology Organization | LTF 100-2017 | Thomas A DeFalco |
| H2020 European Research Council | 639678 | Jürgen Kleine-Vehn |
| Swiss National Science Foundation | 31003A_182625 | Cyril Zipfel |
| Swiss National Science Foundation | 31003A_166577/1 | Christoph Ringli |
| Austrian Science Fund | P 33044 | Jürgen Kleine-Vehn |
| Deutsche Forschungsgemeinschaft | STE 2448/1 | Martin Stegmann |
| Austrian Academy of Sciences | Doctoral fellowship | Kai Dünser |

The funders had no role in study design, data collection and interpretation, or the decision to submit the work for publication.

## Author contributions

Julien Gronnier, Conceptualization, Formal analysis, Funding acquisition, Investigation, Supervision, Visualization, Writing – original draft, Writing – review and editing; Christina M Franck, Martin Stegmann, Thomas A DeFalco, Alicia Abarca, Investigation, Writing – review and editing; Michelle von Arx, Kai Dünser, Investigation; Wenwei Lin, Zhenbiao Yang, Resources; Jürgen Kleine-Vehn, Conceptualization, Funding acquisition, Project administration, Writing – review and editing; Christoph Ringli, Resources, Writing – review and editing; Cyril Zipfel, Conceptualization, Funding acquisition, Project administration, Supervision, Writing – original draft, Writing – review and editing

## Author ORCIDs

Julien Gronnier http://orcid.org/0000-0002-1429-0542
Alicia Abarca http://orcid.org/0000-0003-3569-851X
Jürgen Kleine-Vehn http://orcid.org/0000-0002-4354-3756
Cyril Zipfel http://orcid.org/0000-0003-4935-8583

## Decision letter and Author response

Decision letter https://doi.org/10.7554/eLife.74162.sa1
Author response https://doi.org/10.7554/eLife.74162.sa2

# Additional files

## Supplementary files
- Transparent reporting form
- Source data 1. Raw images of the western blots.

## Data availability

All data generated or analysed during this study are included in the manuscript and supporting files.

The following previously published datasets were used:

| Author(s) | Year | Dataset title | Dataset URL | Database and Identifier |
|---|---|---|---|---|
| The Arabidopsis Genome Initiative | 2000 | Locus: AT3G51550 | https://www.arabidopsis.org/servlets/TairObject?id=36914&type=locus | The Arabidopsis Information Resource, AT3G51550 |

*Continued on next page*

*Continued*

| Author(s) | Year | Dataset title | Dataset URL | Database and Identifier |
|---|---|---|---|---|
| The Arabidopsis Genome Initiative | 2000 | Locus: AT4G13340 | https://www.arabidopsis.org/servlets/TairObject?id=130361&type=locus | The Arabidopsis Information Resource, AT4G13340 |
| The Arabidopsis Genome Initiative | 2000 | Locus: AT3G24480 | https://www.arabidopsis.org/servlets/TairObject?id=39195&type=locus | The Arabidopsis Information Resource, AT3G24480 |
| The Arabidopsis Genome Initiative | 2000 | Locus: AT4G18670 | https://www.arabidopsis.org/servlets/TairObject?id=127900&type=locus | The Arabidopsis Information Resource, AT4G18670 |
| The Arabidopsis Genome Initiative | 2000 | Locus: AT3G16570 | https://www.arabidopsis.org/servlets/TairObject?id=38239&type=locus | The Arabidopsis Information Resource, AT3G16570 |
| The Arabidopsis Genome Initiative | 2000 | Locus: AT5G46330 | https://www.arabidopsis.org/servlets/TairObject?id=134136&type=locus | The Arabidopsis Information Resource, AT5G46330 |
| The Arabidopsis Genome Initiative | 2000 | Locus: AT4G33430 | https://www.arabidopsis.org/servlets/TairObject?id=127207&type=locus | The Arabidopsis Information Resource, AT4G33430 |

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
