## [Editor Report]

In elegant quantitative live-cell imaging and biochemical experiments, the authors show how activity of the plant immune signaling complex FLS2-BAK1 is affected by nanoscale mobility behaviors mediated through peptide signaling and the receptor kinase FERONIA (FER). Additionally, they are able to define separable roles for FER domains in different biological activities. The details of this work advance our understanding of plant immunity, but also provide generalizable concepts about the roles of nanoscale organization in signaling.

---

## [Decision Letter]

**Decision letter after peer review:**

Thank you for submitting your article "Regulation of immune receptor kinases plasma membrane nanoscale landscape by a plant peptide hormone and its receptors" for consideration by *eLife*. Your article has been reviewed by 3 peer reviewers, one of whom is a member of our Board of Reviewing Editors, and the evaluation has been overseen by Jonathan Cooper as the Senior Editor. The following individual involved in review of your submission has agreed to reveal their identity: Jan Petrášek (Reviewer #3).

Essential revisions:

From the written reviews and discussion, there was much enthusiasm for the work, but two general concerns emerged: (1) replicated data were not fully analyzed and reported and (2) the ability of a broad audience to engage with and appreciate the detailed work would be enhanced by modifications to the text. From the individual reviews appended below you will find specific examples of places where modifying text or presentation would have the largest impact.

1) That "similar results were found among three replicates" is mentioned in nearly every figure, but these replicates are not shown. After a healthy discussion among the reviewers about what was essential to show, all three reviewers concurred that in the cases of single particle tracking and the quantitative imaging of mobility (reported, for example, in Figures 1C, G; 2F; 4C), the data from the three replicates should be analyzed and the differences between experiments reported. This is in part because this paper has the potential to serve as a standard bearer for this technique.

In the case of Western blots, showing a single representative blot was fine, but the band intensities should be quantified using standard densitometry scans. We also encourage the authors to include the replicate, uncropped, blots in a data supplement. For the confocal images of leaves in Figure 3C-D, the single set of images is also fine, but additional details about what constitutes a replicate (see comment from Rev 1) is needed. For other phenotypic measurements like seedling fresh weight, reporting the single replicate is fine.

2) A weak experimental point is the examination of protein accumulation on the plasma membrane – which is used to show that the nanodomain results in the stabilization of FLS on the membrane, preventing them from being endocytosed. Here single channel images are used to determine the amount of protein based on fluorescent intensity without further controls (e.g. molecular, biochemical, internal signal controls, to confirm the overall amount of protein in the different lines). Additionally, it is not shown whether fer mutants have altered endocytosis in general. Please either include supporting experiments or modify the text to indicate the limitations of the current study.

3) The manuscript is very detailed as it described the nanoscale localization of receptor kinases, and the jargon and extensive use of abbreviations makes it challenging for readers outside the plant immune signaling world. Additional discussion and presentation of these findings in an integrative model would help to make the details illustrative of general signaling rules and would broaden the impact of this work. In addition, the phrasing of the abstract that includes mention of a "plasma membrane nanoscale landscape" may lead one to expect that paper will focus on the "landscape" of the PM and the structural changes in the PM provoked by the activity of individual receptor kinases. However, the main focus in on the complex dynamics of individual receptor protein kinases with the PM – no less interesting, but not completely aligned with the abstract.

4) Many of the experimental details or choices are not fully explained; for example, FER effects on cell wall integrity are mostly known from root hair work, but the experiments here focus on cotyledon epidermal cells. Please add some rationale for experimental design choices, be clear about what a "sample" or "replicate" entails, and mention potential limitations of the current work.

*Reviewer #1:*

How does a protein identified as having a role in many different and seemingly independent plant responses act in specific ways in those responses? An example in plants is the receptor kinase FERONIA (FER) that has been implicated in numerous processes including cell wall integrity signaling and response to biotic and abiotic stresses. Using a series of cell biological and biochemical approaches, the authors show, using the intensively studied FLS2-BAK1 immune receptor complex and its previous identified ligands and signaling potentiators or inhibitors, that different domains and activities of FER have discrete roles in different cellular events. They further show that FER and LRX proteins may affect the mobility of plasma membrane proteins in their nanodomains, though precisely how loss of FER or LRX leads to changes in nanodomain properties is not entirely clear. This detailed work adds interesting and important information about how individual proteins and cellular milieus can alter signaling. Additional discussion and presentation of these findings is an integrative model would help to make the details illustrative of general signaling rules and would broaden the impact of this work outside of the plant immunity niche.

From the set-up of this paper, I imagined that the authors would show a unified way in which FER and/or LRXs affected a cellular feature (e.g. nanodomain formation) and this would account for multiple activities. The results of their experiments, however, showed that they can separate activities of FER (e.g. malectin A domain being involved in cell morphogenesis but not PTI whereas FER kinase activity is not required for PTI). I am left not quite knowing how direct FER activity is on FLS2-BAK1 behaviors. I was expecting it to be indirect through some nanodomain organization, but I failed to find this thread followed through. The direct effects of FER are also interesting results, but as an outsider to the PTI and cell wall integrity fields, I find it challenging to synthesize the results into a general overall model. I don't think the authors need more experiments, but I do think a figure that summarizes their updated concept of how FER works would make this paper accessible and appreciated by a wider audience.

*Reviewer #2:*

This paper provides insight into the molecular mechanisms which mediate a plants immune response. Specifically looking at the dynamic organization of nano-domains of immune receptors, where the role of hormone peptides, co-receptors and the cell wall upon this domain are examined. This thus provides insights into overall plant physiology and into the finer details of membrane dynamic organization and signaling hubs. The authors use a range of experimental approaches (genetic, biochemical, live imaging) to show that FER regulates the formation of the FLS2-BAK1 complex, that additional receptors also regulate the stabilization these complexes on membrane as 'nano-domains'. Furthermore, they show that a domain of FER that regulates cell shape is dispensable for the immune response and RALF23 (a ligand for FER) also stabilizes BAK into nanodomains.

The story presented shows a logical progression and is based upon the results presented.

The major strength of the paper is the methods used. For example, high resolution imaging of the cell surface is utilized to directly visualize the dynamics of these proteins, and this is coupled with single particle tracking analysis, thus providing an accurate picture of these domains and their dynamics. Genetic lines are used to specifically examine the functionality of the proteins and their domains.

However, in order to improve the manuscript, I believe the authors should provide further details about the methods used and analysis details. For example, at present many of the figures are presented with statistical analysis to compare experimental conditions, but in the figure legends they state 'similar results were obtained in three independent experiments'. To present the data as robustly as possible, I suggest that the authors should show and perform statistical analysis on these similar experiments. Thus, providing the readers with a chance to evaluate how robust the effect is and to understand how variable the experiments were.

A weak experimental point is the examination of protein accumulation on the plasma membrane – which is used to show that the nanodomain results in the stabilization of FLS on the membrane, preventing them from being endocytosed. Here single channel images are used to determine the amount of protein based on fluorescent intensity without further controls (e.g. molecular, biochemical, internal signal controls, to confirm the overall amount of protein in the different lines). Additionally, it is not clarified if the fer mutants used have altered endocytosis in general.

The legends/methods need further details. For example, in figure 1 supplemental 1, it is not clear what the data points are; are they individual tracks, or cells from the same plant? Thus, it is important to clarify further how the analyses was conducted (ie, what the data points plotted are, further details on Ns/repeats).

I am not sure 'propensity' is the appropriate word, perhaps 'property', line 23.

In order to appeal to non-specialists, and to aid the readers comprehension, the authors should consider introducing less abbreviations and focusing on only the critical ones. For example, just in the 1st introduction paragraph (lines 41-55) there are 9, thus at present, it is quite hard to follow the text.

The authors should change the word static to describe the FLS foci on the cell surface (line 108). The foci are not static as they appear and disappear over time, thus they should consider using the terminology, 'laterally stable foci' or something similar.

As much of the analysis of live imaging relies on trackMate, and while the authors detail the settings used, there is no information about how the threshold values was selected. This is important as for example, during the videos, there is bleaching during acquisition which could result in the shortening of tracks. Furthermore, while I understand that is visually easy to show these results with kymographs, the authors should include a histogram of the foci spot lifetimes (as they have already tracked the spots) to more robustly depict the data.

In general, while the videos with tracking are a great addition to the manuscript, at present the fact that the tracks remain after the spots have disappeared is distracting and makes it hard to see the dynamics of the foci. It should be simple enough to change the videos with trackMate (it is just a case of changing the track display mode to 'show local tracks' and play with the 'show track depth' option), which would greatly improve the usefulness of the tracking videos.

The western blots should be quantified to show the results are robust and reproducible. And there are some signals which appear to be saturated.

While I understand the focus is on the FLS BAK dynamics, I think it would be interesting to show how specific this interaction is for mediating the formation of the nanodomains. For example, by examining another receptor or cargo in the mutant lines it would tell us if FER is a general nanodomain scaffold protein.

Line 114 – authors should state how many frames were combined to create the average projections.

There are no scale bars on the kymographs, so it's impossible to know the duration of imaging/tracks/nanodomains.

Line 123 – I think the authors mean formation/composition and not localization.

Line 145 – reference to figure needs updating.

Line 188 – should be mobile rather than labile.

For the figures showing a single track as a model, it would be good in include a scale bar to allow the reader to understand the scale of these diffusions/domains.

Line 200 – 'deleted' should be mutated, truncated or altered.

206 – should be '…can directly monitor the cell wall.'

219 – co-jointly should be rewritten to say, '..and together they relay..'.

Figure 3 – it would be good to quantify these effects to show how reproducible they are. Maybe for cotyledon – a line profile across the image to show the cell is more wavey? And a density for root hairs over a certain length?

If possible, it would be a great addition to the paper to show that dual dynamics of FLS and BAK in the different experimental conditions.

Line 869 – reference to figure needs updating.

*Reviewer #3:*

The manuscript of Gronnier et al. (23-09-2021-RA-*eLife*-74162) brings an original set of data describing the role of FER receptor kinase in the control of PM organization and dynamics of plant immune receptor kinases FLS2 and its co-receptor BAK1. Using an advanced fluorescence microscopy approach combined with biochemistry and molecular biology, authors show how the perception of plant peptide hormone RALF23 triggers a specific changes in the distribution of receptor kinases FLS2 and BAK1 in cells at the surface of cotyledons (epidermis of 5-day-old young seedlings). Moreover, these changes are shown to be regulated also by other other receptors of FER, i.e. LRX3, LRX4 and LRX5. The formation of the immune receptor kinase complexes with downstream signalling are therefore suggested to be under the control of RALF23 peptide hormone through the action of both FER and LRXs. Moreover, the data allowed authors to conclude that the effects on root hair growth and immune response are uncoupled in the case of both FER and LRXs receptor pathways and that the kinase activity of FER is needed for the inhibitory effect of RALF23 on the immune response, while the role of FER kinase in the pattern recognition receptors-triggered immunity is kinase-independent. The manuscript is very detailed, bringing the description of the nanoscale localization of receptor kinases; however, the biology behind the observed effects is still a point for future research. Results are basically supported by data, although I have some comments on their presentation.

Strengths

This work combines in a very effective way advanced fluorescence microscopy, biochemistry and molecular genetics, this all in a well-established model of *Arabidopsis thaliana*. I really appreciate the level of microscopy details that were possible to perform in a quantitative way. For sure, the microscopy approach shown here is very important for any future work in this field and in numerous technical aspects, it truly paves the road for other researchers.

It is very important that there are more elements of the pathway analyzed in one experimental/observation setup. Without considering both FER and LRXs and evaluating them separately, it would not be possible to conclude on the extent of the changes in the nanoscale organization of RKs involved in the RALF23-controlled pathway. Of course, such work is very technology-demanding and time-consuming.

Weaknesses

I feel that this report needs more attention to the biology itself. For the broader community, it would be perfect to understand in what process the mechanism described here is crucial. Therefore, I feel that authors would much improve this manuscript if they would be able to defend why they use epidermal lobed cells in 5-day-old seedlings. I know that there might be plenty of technical reasons, previous work, etc., but biologists would ask about it; considering that effects on cell growth are shown in root hairs, while all immune responses are studied in cotyledon epidermal lobed cells. The introduction on why it is actually so important to study described processes in cotyledons would help.

Perhaps I am wrong, but the "plasma membrane nanoscale landscape", as mentioned in the last sentence of the abstract, is related to the nanoscale organization of receptor kinases studied here, not the "landscape" of the PM itself. Of course, PM is extremely dynamic, but this manuscript is not focused on the understanding of PM structural changes provoked by the activity of individual receptor kinases. It is rather focused on surprisingly complex dynamics of individual receptor protein kinases with the PM. This I feel needs to be presented in a clearer form.

Statistics is provided for the majority of analyses. However, authors mention in numerous cases (at least in 17 analyses) that "similar results were obtained in three independent experiments". I think that in the case of quantifications of microscopy images, it would be perfect to understand how observed differences in the dynamics of receptor kinases are robust when analyzed in these three mentioned biological repetitions. It would also be informative to include some rationale on the selection of cells for the analysis, e.g. was the size the criterion or something else?

For a broader community of readers, it might be perhaps better to introduce a bit what is that „peptide hormone". I know that authors are very deeply involved in the RALF23 and often simply call this molecule „peptide". However, in the title of this manuscript, the term “peptide hormone" is used, but, the word "hormone" is not used in the manuscript at all. For broader community, this is a bit difficult to follow.

I think that for sure the biological implication of this work would be enhanced if data from biological repetitions mentioned in the text would be involved.

Kymographic analyses are not described in methods nor in captions. Axes of kymograms shown in the manuscript are not annotated; therefore it is not clear how actually dynamic the processes are. Time scale would help here.

In the Figure 1, suppl. Figure 1 the caption is not mentioning the statistics used in this analysis.

Line 877 – subscript should be used for numbers in chemical formulas.

VA-TIRFM is mentioned by authors as the main microscopy method used in this contribution. I hope I got it right, therefore, the abbreviation TIRFM in all main and supplementary captions should be changed to VA-TIRFM, as well as in the description of the microscopy itself (lines 807 and 862).

Line 869 – the reference to the suppl. image is not correct, it should not be Sup Figure 15, but Figure Suppl 6.

The quality of language is very good, however, there are some subtle grammar issues, e.g. on line 136 – „BAK1 might dynamically associates with", I found also some typos etc (line 831, the sentence should begin with a capital letter). I did not have time to find all of them, I encourage authors to check it again.

In vivo advanced fluorescence GSD microscopy is used here and I appreciate a lot this technique and how it is implemented. It would be perhaps good to discuss how far individual markers characterize the mobility of the structure where it is located (PM, cell wall, cytoskeleton, etc.) and how far this technique might be taken as the characterization of the mobility of the particular molecule within the particular structure.

---

## [Author Response]

Essential revisions:From the written reviews and discussion, there was much enthusiasm for the work, but two general concerns emerged: (1) replicated data were not fully analyzed and reported and (2) the ability of a broad audience to engage with and appreciate the detailed work would be enhanced by modifications to the text. From the individual reviews appended below you will find specific examples of places where modifying text or presentation would have the largest impact.1) That "similar results were found among three replicates" is mentioned in nearly every figure, but these replicates are not shown. After a healthy discussion among the reviewers about what was essential to show, all three reviewers concurred that in the cases of single particle tracking and the quantitative imaging of mobility (reported, for example, in Figures 1C, G; 2F; 4C), the data from the three replicates should be analyzed and the differences between experiments reported. This is in part because this paper has the potential to serve as a standard bearer for this technique.

We now present data of individual replicate experiment as well as their statistical analysis.

In the case of Western blots, showing a single representative blot was fine, but the band intensities should be quantified using standard densitometry scans. We also encourage the authors to include the replicate, uncropped, blots in a data supplement. For the confocal images of leaves in Figure 3C-D, the single set of images is also fine, but additional details about what constitutes a replicate (see comment from Rev 1) is needed. For other phenotypic measurements like seedling fresh weight, reporting the single replicate is fine.

We have quantified the bands intensities (values shown below each line) of the western blots. We present replicate experiments, and corresponding uncropped blots in supplementary data. We also clarified what constitutes a replicate for Figure 3C and D.

2) A weak experimental point is the examination of protein accumulation on the plasma membrane – which is used to show that the nanodomain results in the stabilization of FLS on the membrane, preventing them from being endocytosed. Here single channel images are used to determine the amount of protein based on fluorescent intensity without further controls (e.g. molecular, biochemical, internal signal controls, to confirm the overall amount of protein in the different lines). Additionally, it is not shown whether fer mutants have altered endocytosis in general. Please either include supporting experiments or modify the text to indicate the limitations of the current study.

We think that the use of single channel images to analyze the amount of tagged proteins is well-suited to quantify protein accumulation at the plasma membrane. Indeed, it has previously been used to robustly quantify FLS2 plasma membrane accumulation (*e.g.* Göhre et al., Curr. Biol. 2008; Smith et al., PLoS Pathog. 2014; Wang et al., New Phytologist 2020), and was shown to reveal defect in protein accumulation that could otherwise only be detected by tedious biochemically purification of the plasma membrane (*e.g.* Collins et al., Plant Physiol. 2020). Furthermore, VA-TIRM experiments consistently showed a decrease in FLS2-GFP accumulation at the plasma membrane (Figure 1; Figure 1 – supplemental figure 2 and Figure 4 – supplemental figure 4). Similar to previous studies, we observed that compromised in nanodomain organization correlates with a defect in plasma membrane accumulation, but we do not claim that these observations demonstrate causality. As suggested by Reviewer 2, we now also discuss the potential involvement of the proposed regulation of endocytosis by FER (Yu et al., Development 2020).

3) The manuscript is very detailed as it described the nanoscale localization of receptor kinases, and the jargon and extensive use of abbreviations makes it challenging for readers outside the plant immune signaling world. Additional discussion and presentation of these findings in an integrative model would help to make the details illustrative of general signaling rules and would broaden the impact of this work. In addition, the phrasing of the abstract that includes mention of a "plasma membrane nanoscale landscape" may lead one to expect that paper will focus on the "landscape" of the PM and the structural changes in the PM provoked by the activity of individual receptor kinases. However, the main focus in on the complex dynamics of individual receptor protein kinases with the PM – no less interesting, but not completely aligned with the abstract.

We reduced the number of abbreviations. We now further discuss our findings and present a model. We agree that the word ‘landscape’ would imply a broader investigation of the properties of the plasma membrane, and therefore replaced landscape by organization in the abstract and in the title.

4) Many of the experimental details or choices are not fully explained; for example, FER effects on cell wall integrity are mostly known from root hair work, but the experiments here focus on cotyledon epidermal cells. Please add some rationale for experimental design choices, be clear about what a "sample" or "replicate" entails, and mention potential limitations of the current work.

We now further explain experimental details, and clarified the meaning of sample and replicate in the figure legends and in the method section.

FERONIA function in cell wall integrity is well described in cotyledon epidermal cells and was linked to cell morphogenesis (e.g. Lin et al., bioRxiv 2018; Lin et al., Curr. Biol. 2021) and mechano-sensing (Malivert et al., PLoS Biol. 2021; Tang et al., Curr. Biol. 2021). We recapitulated observations related to cell morphogenesis in our laboratory conditions (Figure 3 C,D) to provide a clear comparative analysis with the immune signaling outputs.

Reviewer #1:How does a protein identified as having a role in many different and seemingly independent plant responses act in specific ways in those responses? An example in plants is the receptor kinase FERONIA (FER) that has been implicated in numerous processes including cell wall integrity signaling and response to biotic and abiotic stresses. Using a series of cell biological and biochemical approaches, the authors show, using the intensively studied FLS2-BAK1 immune receptor complex and its previous identified ligands and signaling potentiators or inhibitors, that different domains and activities of FER have discrete roles in different cellular events. They further show that FER and LRX proteins may affect the mobility of plasma membrane proteins in their nanodomains, though precisely how loss of FER or LRX leads to changes in nanodomain properties is not entirely clear. This detailed work adds interesting and important information about how individual proteins and cellular milieus can alter signaling. Additional discussion and presentation of these findings is an integrative model would help to make the details illustrative of general signaling rules and would broaden the impact of this work outside of the plant immunity niche.From the set-up of this paper, I imagined that the authors would show a unified way in which FER and/or LRXs affected a cellular feature (e.g. nanodomain formation) and this would account for multiple activities. The results of their experiments, however, showed that they can separate activities of FER (e.g. malectin A domain being involved in cell morphogenesis but not PTI whereas FER kinase activity is not required for PTI). I am left not quite knowing how direct FER activity is on FLS2-BAK1 behaviors. I was expecting it to be indirect through some nanodomain organization, but I failed to find this thread followed through. The direct effects of FER are also interesting results, but as an outsider to the PTI and cell wall integrity fields, I find it challenging to synthesize the results into a general overall model. I don't think the authors need more experiments, but I do think a figure that summarizes their updated concept of how FER works would make this paper accessible and appreciated by a wider audience.

We thank the reviewer for her/his positive comments. We now further discuss our findings and present a model illustrating main findings and potential future directions.

Reviewer #2:[…] In order to improve the manuscript, I believe the authors should provide further details about the methods used and analysis details. For example, at present many of the figures are presented with statistical analysis to compare experimental conditions, but in the figure legends they state 'similar results were obtained in three independent experiments'. To present the data as robustly as possible, I suggest that the authors should show and perform statistical analysis on these similar experiments. Thus, providing the readers with a chance to evaluate how robust the effect is and to understand how variable the experiments were.

The manuscript now includes the data and statistical analysis of the independent experiments.

A weak experimental point is the examination of protein accumulation on the plasma membrane – which is used to show that the nanodomain results in the stabilization of FLS on the membrane, preventing them from being endocytosed. Here single channel images are used to determine the amount of protein based on fluorescent intensity without further controls (e.g. molecular, biochemical, internal signal controls, to confirm the overall amount of protein in the different lines). Additionally, it is not clarified if the fer mutants used have altered endocytosis in general.

Please see our answer to editorial comment #2.

The legends/methods need further details. For example, in figure 1 supplemental 1, it is not clear what the data points are; are they individual tracks, or cells from the same plant? Thus, it is important to clarify further how the analyses was conducted (ie, what the data points plotted are, further details on Ns/repeats).

We now further detail the legends regarding sample size and to what correspond the data points for each graph.

I am not sure 'propensity' is the appropriate word, perhaps 'property', line 23.

We think that propensity better reflect the dynamic aspect of spatial partitioning.

In order to appeal to non-specialists, and to aid the readers comprehension, the authors should consider introducing less abbreviations and focusing on only the critical ones. For example, just in the 1st introduction paragraph (lines 41-55) there are 9, thus at present, it is quite hard to follow the text.

We have reduced the number of abbreviations and present a table summarizing them.

The authors should change the word static to describe the FLS foci on the cell surface (line 108). The foci are not static as they appear and disappear over time, thus they should consider using the terminology, 'laterally stable foci' or something similar.

FLS2 foci are now described as laterally stable foci.

As much of the analysis of live imaging relies on trackMate, and while the authors detail the settings used, there is no information about how the threshold values was selected. This is important as for example, during the videos, there is bleaching during acquisition which could result in the shortening of tracks. Furthermore, while I understand that is visually easy to show these results with kymographs, the authors should include a histogram of the foci spot lifetimes (as they have already tracked the spots) to more robustly depict the data.

The thresholds were empirically determined for optimal single particle detection and linking. We do not consider foci spot lifetimes as a relevant metric to describe the lateral organization/dynamics of particles, which is the focus of our study.

In general, while the videos with tracking are a great addition to the manuscript, at present the fact that the tracks remain after the spots have disappeared is distracting and makes it hard to see the dynamics of the foci. It should be simple enough to change the videos with trackMate (it is just a case of changing the track display mode to 'show local tracks' and play with the 'show track depth' option), which would greatly improve the usefulness of the tracking videos.

To the contrary, we think that full tracks better reflect the behavior of the single particles.

The western blots should be quantified to show the results are robust and reproducible. And there are some signals which appear to be saturated.

We now provide quantification of the western blots.

While I understand the focus is on the FLS BAK dynamics, I think it would be interesting to show how specific this interaction is for mediating the formation of the nanodomains. For example, by examining another receptor or cargo in the mutant lines it would tell us if FER is a general nanodomain scaffold protein.

We share the interest of the reviewer, and are actively working to determine whether the FER function described in the present manuscript applies to for additional RK-mediated signaling pathways. However, such detailed investigation is far reaching and beyond the scope of our current manuscript.

Line 114 – authors should state how many frames were combined to create the average projections.

This information is now indicated in each figure legend.

There are no scale bars on the kymographs, so it's impossible to know the duration of imaging/tracks/nanodomains.

We now provide scale bars for the kymographs.

Line 123 – I think the authors mean formation/composition and not localization.

We changed localization to organization. Thank you.

Line 145 – reference to figure needs updating.

This is now corrected. Thank you.

Line 188 – should be mobile rather than labile.

We assumed that the reviewer refers to L118, and changed labile to mobile.

For the figures showing a single track as a model, it would be good in include a scale bar to allow the reader to understand the scale of these diffusions/domains.

These models correspond to graphical illustration summarizing our observations but not to experimental single particle trajectories. We prefer not to include a scale bar, as it may be confusing.

Line 200 – 'deleted' should be mutated, truncated or altered.

This is now corrected. Thank you.

206 – should be '…can directly monitor the cell wall.'

We are here unsure which sentence the reviewer refers to.

219 – co-jointly should be rewritten to say, '..and together they relay..'.

We are here unsure which sentence the reviewer refers to.

Figure 3 – it would be good to quantify these effects to show how reproducible they are. Maybe for cotyledon – a line profile across the image to show the cell is more wavey? And a density for root hairs over a certain length?

We think that these observations are already very clear and are certainly reproductible.

If possible, it would be a great addition to the paper to show that dual dynamics of FLS and BAK in the different experimental conditions.

This remains technically challenging and would require further extensive work.

Line 869 – reference to figure needs updating.

This is corrected, thank you.

Reviewer #3:[…] I feel that this report needs more attention to the biology itself. For the broader community, it would be perfect to understand in what process the mechanism described here is crucial. Therefore, I feel that authors would much improve this manuscript if they would be able to defend why they use epidermal lobed cells in 5-day-old seedlings. I know that there might be plenty of technical reasons, previous work, etc., but biologists would ask about it; considering that effects on cell growth are shown in root hairs, while all immune responses are studied in cotyledon epidermal lobed cells. The introduction on why it is actually so important to study described processes in cotyledons would help.

Please see our answer to the editorial comment #4

Perhaps I am wrong, but the "plasma membrane nanoscale landscape", as mentioned in the last sentence of the abstract, is related to the nanoscale organization of receptor kinases studied here, not the "landscape" of the PM itself. Of course, PM is extremely dynamic, but this manuscript is not focused on the understanding of PM structural changes provoked by the activity of individual receptor kinases. It is rather focused on surprisingly complex dynamics of individual receptor protein kinases with the PM. This I feel needs to be presented in a clearer form.

We agree and have replaced landscape by organization in the abstract.

Statistics is provided for the majority of analyses. However, authors mention in numerous cases (at least in 17 analyses) that "similar results were obtained in three independent experiments". I think that in the case of quantifications of microscopy images, it would be perfect to understand how observed differences in the dynamics of receptor kinases are robust when analyzed in these three mentioned biological repetitions. It would also be informative to include some rationale on the selection of cells for the analysis, e.g. was the size the criterion or something else?

We thank the reviewer for her/his proposition and now provide data from individual experiments and their statistical analysis. Concerning the observation by VA-TIRFM: after placing the center of cotyledons of 5-day-old seedlings (entire seedlings were mounted between cover slips and slides) at the center of the objective lens, we screened, by moving in x and y, using pre-defined and fixed laser angle and power, for cells surfaces in distance range of the evanescent wave without additional a priori selection criterions.

For a broader community of readers, it might be perhaps better to introduce a bit what is that „peptide hormone". I know that authors are very deeply involved in the RALF23 and often simply call this molecule „peptide". However, in the title of this manuscript, the term “peptide hormone" is used, but, the word "hormone" is not used in the manuscript at all. For broader community, this is a bit difficult to follow.I think that for sure the biological implication of this work would be enhanced if data from biological repetitions mentioned in the text would be involved.

We now introduce RALF23 as a peptide hormone in the introduction. Thank you.

Kymographic analyses are not described in methods nor in captions. Axes of kymograms shown in the manuscript are not annotated; therefore it is not clear how actually dynamic the processes are. Time scale would help here.

We have added scale bars for all kymographs. The kymographs were generated using “orthogonal views” (Image/Stacks/Orthogonal views). This information has been added to the material and methods section. Thank you.

In the Figure 1, suppl. Figure 1 the caption is not mentioning the statistics used in this analysis.

This is now implemented. Thank you.

Line 877 – subscript should be used for numbers in chemical formulas.

This is now corrected. Thank you.

VA-TIRFM is mentioned by authors as the main microscopy method used in this contribution. I hope I got it right, therefore, the abbreviation TIRFM in all main and supplementary captions should be changed to VA-TIRFM, as well as in the description of the microscopy itself (lines 807 and 862).

This is now corrected; we replaced TIRFM by VA-TIRFM throughout the text.

Line 869 – the reference to the suppl. image is not correct, it should not be Sup Figure 15, but Figure Suppl 6.

This is now corrected. Thank you.

The quality of language is very good, however, there are some subtle grammar issues, e.g. on line 136 – „BAK1 might dynamically associates with", I found also some typos etc (line 831, the sentence should begin with a capital letter). I did not have time to find all of them, I encourage authors to check it again.

We have now checked and corrected the manuscript for grammar issues. Thank you.

In vivo advanced fluorescence GSD microscopy is used here and I appreciate a lot this technique and how it is implemented. It would be perhaps good to discuss how far individual markers characterize the mobility of the structure where it is located (PM, cell wall, cytoskeleton, etc.) and how far this technique might be taken as the characterization of the mobility of the particular molecule within the particular structure.

Thank you for your interest. We haven’t performed such a comparative analysis and are therefore unfortunately not in the position of evaluating it limits.